# Reinforcement Learning with a Terminator

**Guy Tennenholtz** [*][†]
guytenn@gmail.com

**Nadav Merlis** [*]

**Lior Shani** [*]

**Shie Mannor** [*][†]

**Uri Shalit** [*]

**Gal Chechik** [†][‡]

**Assaf Hallak** [†]

**Gal Dalal** [†]

## Abstract

We present the problem of reinforcement learning with exogenous termination. We define the Termination Markov Decision Process (TerMDP), an extension of the MDP framework, in which episodes may be interrupted by an external non-Markovian observer. This formulation accounts for numerous real-world situations, such as a human interrupting an autonomous driving agent for reasons of discomfort. We learn the parameters of the TerMDP and leverage the structure of the estimation problem to provide state-wise confidence bounds. We use these to construct a provably-efficient algorithm, which accounts for termination, and bound its regret. Motivated by our theoretical analysis, we design and implement a scalable approach, which combines optimism (w.r.t. termination) and a dynamic discount factor, incorporating the termination probability. We deploy our method on high-dimensional driving and MinAtar benchmarks. Additionally, we test our approach on human data in a driving setting. Our results demonstrate fast convergence and significant improvement over various baseline approaches.

## 1  Introduction

The field of reinforcement learning (RL) involves an agent interacting with an environment, maximizing a cumulative reward [Puterman, 2014]. As RL becomes more instrumental in real-world applications [Lazic et al., 2018, Kiran et al., 2021, Mandhane et al., 2022], exogenous inputs beyond the prespecified reward pose a new challenge. Particularly, an external authority (e.g., a human operator) may decide to terminate the agent's operation when it detects undesirable behavior. In this work, we generalize the basic RL framework to accommodate such external feedback.

We propose a generalization of the standard Markov Decision Process (MDP), in which external termination can occur due to a non-Markovian observer. When terminated, the agent stops interacting with the environment and cannot collect additional rewards. This setup describes various real-world scenarios, including: passengers in autonomous vehicles [Le Vine et al., 2015, Zhu et al., 2020], users in recommender systems [Wang et al., 2009], employees terminating their contracts (churn management) [Sisodia et al., 2017], and operators in factories; particularly, datacenter cooling systems, or other safety-critical systems, which require constant monitoring and rare, though critical, human takeovers [Modares et al., 2015]. In these tasks, human preferences, incentives, and constraints play a central role, and designing a reward function to capture them may be highly complex. Instead, we propose to let the agent itself learn these latent human utilities by leveraging the termination events.

We introduce the Termination Markov Decision Process (TerMDP), depicted in Figure 1. We consider a terminator, observing the agent, which aggregates penalties w.r.t. a predetermined, state-action-dependent, yet *unknown*, cost function. As the agent progresses, unfavorable states accumulate costs

---

[*]Technion, Israel institute of technology
[†]Nvidia Research, Israel
[‡]Bar Ilan University, Israel

36th Conference on Neural Information Processing Systems (NeurIPS 2022).

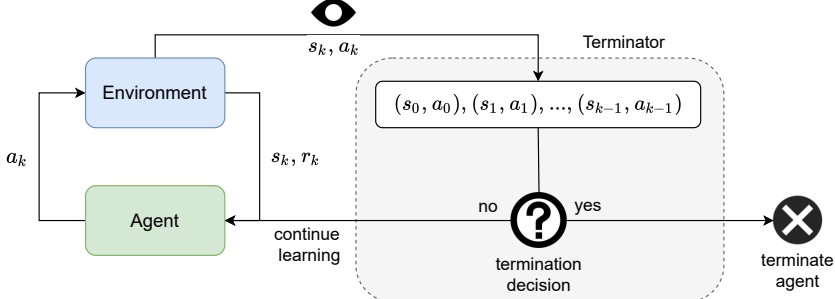

Figure 1: A block diagram of the TerMDP framework. An agent interacts with an environment while an exogenous observer (i.e., terminator) can choose to terminate the agent based on previous interactions. If the agent is terminated, it transitions to a sink state where a reward of 0 is given until the end of the episode.

that gradually increase the terminator's inclination to stop the agent and end the current episode. Receiving merely the sparse termination signals, the agent must learn to behave in the environment, adhering to the terminator's preferences while maximizing reward.

Our contributions are as follows. **(1)** We introduce a novel history-dependent termination model, a natural extension of the MDP framework which incorporates non-trivial termination (Section 2). **(2)** We learn the unknown costs from the implicit termination feedback (Section 3), and provide local guarantees w.r.t. every visited state. We leverage our results to construct a tractable algorithm and provide regret guarantees. **(3)** Building upon our theoretical results, we devise a practical approach that combines optimism with a cost-dependent discount factor, which we test on MinAtar [Young and Tian, 2019] and a new driving benchmark. **(4)** We demonstrate the efficiency of our method on these benchmarks as well as on human-collected termination data (Section 5). Our results show significant improvement over other candidate solutions, which involve direct termination penalties and history-dependent approaches. We also introduce a new task for RL – a driving simulation game which can be easily deployed on mobile phones, consoles, and PC [4].

## 2   Termination Markov Decision Process

We begin by presenting the termination framework and the notation used throughout the paper. Informally, we model the termination problem using a logistic model of past "bad behaviors". We use an unobserved state-action-dependent cost function to capture these external preferences. As the overall cost increases throughout time, so does the probability of termination.

For a positive integer $n$, we denote $[n] = \{1, \ldots, n\}$. We define the Termination Markov Decision Process (TerMDP) by the tuple $\mathcal{M}_T = (\mathcal{S}, \mathcal{A}, P, R, H, c)$, where $\mathcal{S}$ and $\mathcal{A}$ are state and action spaces with cardinality $S$ and $A$, respectively, and $H \in \mathbb{N}$ is the maximal horizon. We consider the following protocol, which proceeds in discrete episodes $k = 1, 2, \ldots, K$. At the beginning of each episode $k$, an agent is initialized at state $s_1^k \in \mathcal{S}$. At every time step $h$ of episode $k$, the agent is at state $s_h^k \in \mathcal{S}$, takes an action $a_h^k \in \mathcal{A}$ and receives a random reward $R_h^k \in [0,1]$ generated from a fixed distribution with mean $r_h(s_h^k, a_h^k)$. A terminator overseeing the agent utilizes a cost function $c : [H] \times \mathcal{S} \times \mathcal{A} \mapsto \mathbb{R}$ that is unobserved and *unknown to the agent*. At time step $h$, the episode terminates with probability

$$\rho_h^k(c) = \rho \left( \sum_{t=1}^h c_t(s_t^k, a_t^k) - b \right),$$

where $\rho(x) = (1 + \exp(-x))^{-1}$ is the logistic function and $b \in \mathbb{R}$ is a bias term which determines the termination probability when no costs are aggregated. Upon termination, the agent transitions to a terminal state $s_{\text{term}}$ which yields no reward, i.e., $r_h(s_{\text{term}}, a) = 0$ for all $h \in [H], a \in \mathcal{A}$. If no termination occurs, the agent transitions to a next state $s_{h+1}^k$ with probability $P_h(s_{h+1}^k | s_h^k, a_h^k)$. Let $t_k^* = \min\{h : s_h^k = s_{\text{term}}\} - 1$ be the time step when the $k^{\text{th}}$ episode was terminated. Notice that the termination probability is non-Markovian, as it depends on the entire trajectory history. We also

---

[4]Code for Backseat Driver and our method, TermPG, can be found at https://github.com/guytenn/Terminator.

note that, when $c \equiv 0$, the TerMDP reduces to a finite horizon MDP with discount factor $\gamma = \rho(-b)$. Finally, we note that our model allows for negative costs. Indeed, these may capture satisfactory behavior, diminishing the effect of previous mistakes, and decreasing the probability of termination.

We define a stochastic, history dependent policy $\pi_h(s_h, \tau_{1:h})$ which maps trajectories $\tau_{1:h} = (s_1, a_1, \ldots, s_{h-1}, a_{h-1})$ up to time step $h$ (excluding) and the $h^{\text{th}}$ states $s_h$ to probability distributions over $\mathcal{A}$. Its value is defined by $V_h^\pi(s, \tau) = \mathbb{E}\left[ \sum_{t=h}^H r_t(s_t, a_t) \mid s_h = s, \tau_{1:h} = \tau, a_t \sim \pi_t(s_t, \tau_{1:t}) \right]$. With slight abuse of notation, we denote the value at the initial time step by $V_1^\pi(s)$. An optimal policy $\pi^*$ maximizes the value for all states and histories simultaneously [5]; we denote its value function by $V^*$. We measure the performance of an agent by its *regret*; namely, the difference between the cumulative value it achieves and the value of an optimal policy,

$$\text{Reg}(K) = \sum_{k=1}^K V_1^*(s_1^k) - V_1^{\pi^k}(s_1^k).$$

**Notations.** We denote the Euclidean norm by $\|\cdot\|_2$ and the Mahalanobis norm induced by the positive definite matrix $A \succ 0$ by $\|x\|_A = \sqrt{x^T A x}$. We denote by $n_h^k(s, a)$ the number of times that a state action pair $(s, a)$ was visited at the $h^{\text{th}}$ time step before the $k^{\text{th}}$ episode. Similarly, we denote by $\hat{X}_h^k(s, a)$ the empirical average of a random variable $X$ (e.g., reward and transition kernel) at $(s, a)$ in the $h^{\text{th}}$ time step, based on all samples before the $k^{\text{th}}$ episode.

We assume there exists a known constant $L$ that bounds the norm of the costs; namely, $\sqrt{\sum_{s,a} \sum_{t=1}^H c_t^2(s_t, a_t)} \leq L$, and denote the set of possible costs by $\mathcal{C}$. We also denote the maximal reciprocal derivative of the logistic function by $\kappa = \max_{h \in [H]} \max_{\{(s_t, a_t)\}_{t=1}^h \in (\mathcal{S} \times \mathcal{A})^h} \left( \dot{\rho}\left( \sum_{t=1}^h c_t(s_t, a_t) - b \right) \right)^{-1}$. This factor will be evident in our theoretical analysis in the next section, as estimating the costs in regions of saturation of the sigmoid is more difficult when the derivative nears zero. Finally, we use $\mathcal{O}(x)$ to refer to a quantity that depends on $x$ up to a poly-log expression in $S, A, K, H, L, \kappa$ and $\log\left(\frac{1}{\delta}\right)$.

## 3 An Optimistic Approach to Overcoming Termination

Unlike the standard MDP setup, in the TerMDP model, the agent can potentially be terminated at any time step. Consider the TerMDP model for which the costs are *known*. We can define a Markov policy $\pi_h$ mapping augmented states $\mathcal{S} \times \mathbb{R}$ to a probability distribution over actions, where here, the state space is augmented by the accumulated costs $\sum_{t=1}^{h-1} c_t(s_t, a_t)$. There exists a policy, which does not use historical information, besides the accumulated costs, and achieves the value of the optimal history-dependent policy (see Appendix C). Therefore, when solving for an optimal policy (e.g., by planning), one can use the current accumulated cost instead of the full trajectory history.

This suggests a plausible approach for solving the TerMDP – first learn the cost function, and then solve the state-augmented MDP for which the costs are known. This, in turn, leads to the following question: **can we learn the costs $c$ from the termination signals?** In what follows, we answer this question affirmatively. We show that by using the termination structure, one can efficiently converge to the true cost function *locally* – for every state and action. We provide uncertainty estimates for the state-wise costs, which allow us to construct an efficient optimistic algorithm for solving the problem.

**Learning the Costs.** To learn the costs, we show that the agent can effectively gain information about costs even in time steps where no termination occurs. Recall that at any time step $h \in [H-1]$, the agent acquires a sample from a Bernoulli random variable with parameter $p = \rho_h^k(c) = \rho\left( \sum_{t=1}^h c_t(s_t^k, a_t^k) - b \right)$. Notably, a lack of termination, which occurs with probability $1 - \rho_h^k(c)$, is also an informative signal of the unknown costs. We propose to leverage this information by recognizing the costs $c$ as parameters of a probabilistic model, maximizing their likelihood. We use

---

[5]Such a policy always exists; we can always augment the state space with the history, which would make the environment Markovian and imply the existence of an optimal history-dependent policy [Puterman, 2014].

---

**Algorithm 1** TermCRL: Termination Confidence Reinforcement Learning

---
1: **require:** $\lambda > 0$
2: **for** $k = 1, \ldots, K$ **do**
3:     **for** $(h, s, a) \in [H] \times \mathcal{S} \times \mathcal{A}$ **do**
4:         $\bar{r}_h^k(s, a) = \hat{r}_h^k(s, a) + b_k^r(h, s, a) + b_k^p(h, s, a)$
5:         $\bar{c}_h^k(s, a) = \hat{c}_h^k(s, a) - b_k^c(h, s, a)$                       // Appendix I.1
6:     **end for**
7:     $\pi^k \leftarrow \text{TerMDP-Plan}\Big(\mathcal{M}_T\big(\mathcal{S}, \mathcal{A}, H, \bar{r}^k, \hat{P}^k, \bar{c}^k\big)\Big)$         // Appendix H
8:     Rollout a trajectory by acting $\pi^k$
9:     $\hat{c}^{k+1} \in \arg\max_{c \in \mathcal{C}} \mathcal{L}_\lambda^k(c)$                              // Equation (1)
10:    Update $\hat{P}^{k+1}(s, a), \hat{r}^{k+1}(s, a), n^{k+1}(s, a)$ over rollout trajectory
11: **end for**

---

the regularized cross-entropy, defined for some $\lambda > 0$ by

$$\mathcal{L}_\lambda^k(c) = \sum_{k'=1}^{k} \sum_{h=1}^{H-1} \left[ \mathbb{1}\{h < t_{k'}^*\} \log\big(1 - \rho_h^k(c)\big) + \mathbb{1}\{h = t_{k'}^*\} \log\big(\rho_h^k(c)\big) \right] - \lambda \|c\|_2^2. \quad (1)$$

By maximizing the cost likelihood in Equation (1), global guarantees of the cost can be achieved, similar to previous work on logistic bandits [Zhang et al., 2016, Abeille et al., 2021]. Particularly, denoting by $\hat{c}^k \in \arg\max \mathcal{L}_\lambda^k(c)$ the maximum likelihood estimates of the costs, it can be shown that for any history, a global upper bound on $\|\hat{c}^k - c\|_{\Sigma_k}$ can be obtained, where the history-dependent design matrix $\Sigma_k$ captures the empirical correlations of visitation frequencies (see Appendix K for details). Unfortunately, using $\|\hat{c}^k - c\|_{\Sigma_k}$ amounts to an intractable algorithm [Chatterji et al., 2021], and thus to an undesirable result.

Instead, as terminations are sampled on *every time step* (i.e., non-terminations are informative signals as well), we show we can obtain a *local* bound on the cost function $c$. Specifically, we show that the error $|\hat{c}_h^k(s, a) - c_h(s, a)|$ diminishes with $n_h^k(s, a)$. The following result is a main contribution of our work, and the crux of our regret guarantees later on (see Appendix K for proof).

**Theorem 1** (Local Cost Estimation Confidence Bound). *Let $\hat{c}^k \in \arg\max_{c \in \mathcal{C}} \mathcal{L}_\lambda^k(c)$ be the maximum likelihood estimate of the costs. Then, for any $\delta > 0$, with probability of at least $1 - \delta$, for all episodes $k \in [K]$, timesteps $h \in [H - 1]$ and state-actions $(s, a) \in \mathcal{S} \times \mathcal{A}$, it holds that*

$$\left|\hat{c}_h^k(s, a) - c_h(s, a)\right| \leq \mathcal{O}\left( \big(n_h^k(s, a)\big)^{-0.5} \sqrt{\kappa S A H L^3} \log\left(\frac{1}{\delta}\left(1 + \frac{kL}{S^2 A^2 H}\right)\right)\right).$$

We note the presence of $\kappa$ in our upper bound, a common factor [Chatterji et al., 2021], which is fundamental to our analysis, capturing the complexity of estimating the costs. Trajectories that saturate the logistic function lead to more difficult credit assignment. Specifically, when the accumulated costs are high, any additional penalty would only marginally change the termination probability, making its estimation harder. A similar argument can be made when the termination probability is low.

We emphasize that in contrast to previous work on global reward feedback in RL [Chatterji et al., 2021, Efroni et al., 2020b], which focused specifically on settings in which information is provided only at the end of an episode, the TerMDP framework provides us with additional information whenever no termination occurs, allowing us to achieve strong, local bounds of the unknown costs. This observation is crucial for the design of a computationally tractable algorithm, as we will see both in theory as well as in our experiments later on.

### 3.1 Termination Confidence Reinforcement Learning

We are now ready to present our method for solving TerMDPs with unknown costs. Our proposed approach, which we call Termination Confidence Reinforcement Learning (TermCRL), is shown in Algorithm 1. Leveraging the local convergence guarantees of Theorem 1, we estimate the costs by maximizing the likelihood in Equation (1). We compensate for uncertainty in the

---

**Algorithm 2** TermPG

---

1: **require:** window $w$, number of ensembles $M$, number of rollouts $N$, number of iterations $K$, policy gradient algorithm `ALG-PG`

2: **initialize:** $\mathcal{B}_{\text{pos}} \leftarrow \emptyset, \mathcal{B}_{\text{neg}} \leftarrow \emptyset, \pi_\theta \leftarrow$ random initialization

3: **for** $k = 1, \ldots, K$ **do**

4:     Rollout $N$ trajectories using $\pi_\theta$, $\mathcal{R} = \left\{ s_1^i, a_1^i, r_1^i, \ldots, s_{t_i^*}^i, a_{t_i^*}^i, r_{t_i^*}^i \right\}_{i=1}^N$.

5:     **for** $i = 1, \ldots, N$ **do**

6:         Add $t_i^* - 1$ negative examples $\left( s_{\max\{1,l-w+1\}}, a_{\max\{1,l-w+1\}}, \ldots, s_l, a_l \right)_{l=1}^{t^*-1}$ to $\mathcal{B}_{\text{neg}}$.

7:         Add one positive example $\left( s_{\max\{1,t^*-w+1\}}, a_{t^*-\max\{1,t^*-w+1\}}, \ldots, s_{t^*}, a_{t^*} \right)$.

8:     **end for**

9:     Train bootstrap ensemble $\{c_{\phi_m}\}_{m=1}^M$ using binary cross entropy over data $\mathcal{B}_{\text{neg}}, \mathcal{B}_{\text{pos}}$.

10:     Augment states in $\mathcal{R}$ by $s_l^i \leftarrow s_l^i \cup \sum_{j=1}^{\min\{w,l\}} \min_m c_{\phi_m}(s_{l-j}^i, a_{l-j}^i)$.

11:     Update policy $\pi_\theta \leftarrow$ `ALG-PG`$(\mathcal{R})$ with dynamic discount (see Section 4.2).

12: **end for**

---

reward, transitions, and costs by incorporating optimism. We define bonuses for the reward, transition, and cost function by $b_k^r(h, s, a) = \mathcal{O}\left( \sqrt{\frac{\log(1/\delta)}{n_h^k(s,a) \vee 1}} \right), b_k^p(h, s, a) = \mathcal{O}\left( \sqrt{\frac{SH^2 \log(1/\delta)}{n_h^k(s,a) \vee 1}} \right)$, and $b_k^c(h, s, a) = \mathcal{O}\left( \sqrt{\frac{\kappa SAHL^3}{n_h^k(s,a) \vee 1}} \log\left(\frac{1}{\delta}\right) \right)$ for some $\delta > 0$ (see Appendix I.1 for explicit definitions).

We add the reward and transition bonuses to the estimated reward (line 4), while the optimistic cost bonus is applied directly to the estimated costs (line 5). Then, a planner (line 7) solves the optimistic MDP for which the costs are known and are given by their optimistic counterparts. We refer the reader to Appendix H for further discussion on planning in TerMDPs. The following theorem provides regret guarantees for Algorithm 1. Its proof is given in Appendix J and relies on Theorem 1 and the analysis of UCRL [Auer et al., 2008, Efroni et al., 2019].

**Theorem 2.** *[Regret of TermCRL] With probability at least $1 - \delta$, the regret of Algorithm 1 is*

$$\text{Reg}(K) \leq \mathcal{O}\left( \sqrt{\kappa S^2 A^2 H^{8.5} L^3 K \log^3\left(\frac{SAHK}{\delta}\right)} \right).$$

Compared to the standard regret of UCRL [Auer et al., 2008], an additional $\sqrt{\kappa AH^4 L^3}$ multiplicative factor is evident in our result, which is due to the convergence rates of the costs in Theorem 1. Motivated by our theoretical results, in what follows we propose a practical approach, inspired by Algorithm 1, which utilizes local cost confidence intervals in a deep RL framework.

## 4 Termination Policy Gradient

Following the theoretical analysis in the previous section, we propose a practical approach for solving TerMDPs. Particularly, in this section, we devise a policy gradient method that accounts for the unknown costs leading to termination. We assume a stationary setup for which the transitions, rewards, costs, and policy are time-homogeneous. Our approach consists of three key elements: learning the costs, leveraging uncertainty estimates over costs, and constructing efficient value estimates through a dynamic cost-dependent discount factor.

Algorithm 2 describes the Termination Policy Gradient (TermPG) method, which trains an ensemble of cost networks (to estimate the costs and uncertainty) over rollouts in a policy gradient framework. We represent our policy and cost networks using neural networks with parameters $\theta, \{\phi_m\}_{m=1}^M$. At every iteration, the agent rolls out $N$ trajectories in the environment using a parametric policy, $\pi_\theta$. The rollouts are split into subtrajectories which are labeled w.r.t. the termination signal, where positive labels are used for examples that end with termination. Particularly, we split the rollouts into "windows" (i.e., subtrajectories of length $w$), where a rollout of length $t^*$, which ends with termination, is split into $t^* - 1$ negative examples $\left( s_{\max\{1,l-w+1\}}, a_{\max\{1,l-w+1\}}, \ldots, s_l, a_l \right)_{l=1}^{t^*-1}$,

Figure 2: Block diagram of the cost training procedure. Rollouts are split into subtrajectories, labeled according to whether they end in termination. Given the dataset of labeled subtrajectories, an ensemble of $M$ cost networks is trained end-to-end using cross-entropy with bootstrap samples (all time steps share the same ensemble).

and one positive example $\left(s_{\max\{1,t^*-w+1\}}, a_{t^*-\max\{1,t^*-w+1\}}, \ldots, s_{t^*}, a_{t^*}\right)$. Similarly, a rollout of length $H$ which does not end with termination contains $H$ negative examples. We note that by taking finite windows, we assume the terminator "forgets" accumulated costs that are not recent - a generalization of the theoretical TerMDP model in Section 2, for which $w = H$. In Section 5, we provide experiments of misspecification of the true underlying window width, where this model assumption does not hold.

### 4.1 Learning the Costs

Having collected a dataset of positive and negative examples, we train a logistic regression model consisting of an ensemble of $M$ cost networks $\{c_{\phi_m}\}_{m=1}^M$, shared across timesteps, as depicted in Figure 2. Specifically, for an example $\left(s_{\max\{1,l-w+1\}}, a_{\max\{1,l-w+1\}}, \ldots, s_l, a_l\right)$ we estimate the termination probability by $\rho\left(\sum_{j=1}^{\min\{w,l\}} c_{\phi_m}(s_{l-j+1}, a_{l-j+1}) - b_m\right)$, where $\{b_m\}_{m=1}^M$ are learnable bias parameters. The parameters are then learned end-to-end using the cross entropy loss. We use the bootstrap method [Bickel and Freedman, 1981, Chua et al., 2018] over the ensemble of cost networks. This ensemble is later used in Algorithm 2 to produce optimistic estimates of the costs. Particularly, the agent policy $\pi_\theta$ uses the current state augmented by the optimistic cummulative predicted cost, i.e., $s_l^{\text{aug}} = (s_l, C_{\text{optimistic}})$, where $C_{\text{optimistic}} = \sum_{j=1}^{\min\{w,l\}} \min_m c_{\phi_m}(s_{l-j+1}, a_{l-j+1})$. Finally, the agent is trained with the augmented states using a policy gradient algorithm `ALG-PG` (e.g., PPO [Schulman et al., 2017], IMPALA [Espeholt et al., 2018]).

### 4.2 Optimistic Dynamic Discount Factor

While augmenting the state with the optimistic accumulated costs is sufficient for obtaining optimality, we propose to further leverage these estimates more explicitly – noticing that the finite horizon objective we are solving can be cast to a discounted problem. Particularly, it is well known that the discount factor $\gamma \in (0, 1)$ can be equivalently formulated as the probability of "staying alive" (see the discounted MDP framework, Puterman [2014]). Similarly, by augmenting the state $s$ with the accumulated cost $C_h = \sum_{t=1}^h c(s_t, a_t)$, we view the probability $1 - \rho(C_h)$ as a state-dependent discount factor, capturing the probability of an agent in a TerMDP to not be terminated.

We define a dynamic, cost-dependent discount factor for value estimation. We use the state-action value function $Q(s, a, C)$ over the augmented states, defined for any $s, a, C$ by

$$Q^\pi(s, a, C) = \mathbb{E}_\pi\left[\sum_{t=1}^H \left(\prod_{h=1}^t \gamma_h\right) r(s_t, a_t) \,\middle|\, s_1 = s, a_1 = a, C_1 = C\right],$$

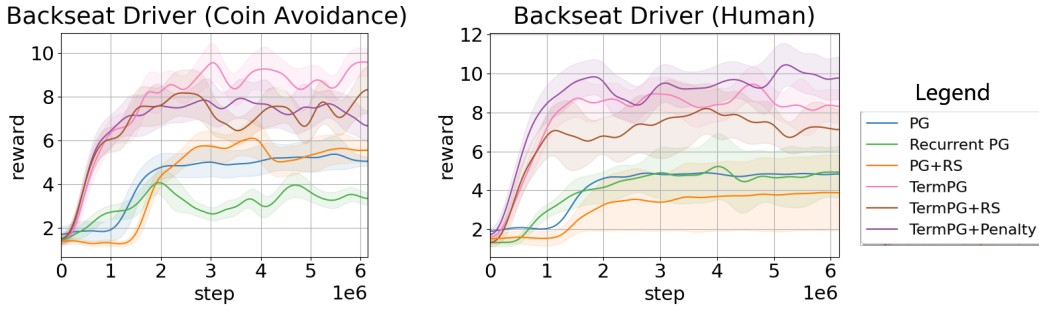

Figure 3: Mean reward with std. over five seeds of "Backseat Driver". Left: coin avoidance; right: human termination. Variants with reward shaping (RS, orange and brown) penalize the agent with a constant value upon termination. The recurrent PG variant (green) uses a history-dependent policy without learning costs. The TermPG+Penalty variant (purple) penalizes the reward at every time step using the estimated costs.

where $\gamma_h = 1 - \rho\left(C + \sum_{i=2}^{h-1} c(s_i, a_i) - b\right)$. This yields the Termination Bellman Equations (see Appendix D for derivation)

$$Q^\pi(s, a, C) = r(s, a) + (1 - \rho(C))\mathbb{E}_{s' \sim P(\cdot|s,a), a' \sim \pi(s')}[Q^\pi(s', a', C + c(s', a'))].$$

To incorporate uncertainty in the estimated costs, we use the optimistic accumulated costs $C_{\text{optimistic}} = \sum_{j=1}^{\min\{w,l\}} \min_m c_{\phi_m}(s_{l-j+1}, a_{l-j+1})$. Then, the discount factor becomes $\gamma(C_{\text{optimistic}}) = 1 - \rho(C_{\text{optimistic}} - b)$. Assuming that, w.h.p., optimistic costs are smaller than the true costs, the discount factor decreases as the agent exploits previously visited states.

The dynamic discount factor allows us to obtain a more accurate value estimator. In particular, we leverage the optimistic cost-dependent discount factor $\gamma(C_{\text{optimistic}})$ in our value estimation procedure, using Generalized Advantage Estimation (GAE, Schulman et al. [2015]). As we will show in the next section, using the optimistic discount factor significantly improves overall performance.

## 5 Experiments

In this section we evaluate the strength of our approach, comparing it to several baselines, including: **(1) PG (naive):** The standard policy gradient without additional assumptions, which ignores termination. **(2) Recurrent PG:** The standard policy gradient with a history-dependent recurrent policy (without cost estimation or dynamic discount factor). As the history is a sufficient statistic of the costs, the optimal policy is realizable. **(3) PG with Reward Shaping (RS):** We penalize the reward upon termination by a constant value, i.e., $r(s, a) - p\mathbb{1}\{s_{\text{term}}\}$, for some $p > 0$. This approach can be applied to any variant of Algorithm 2 or the methods listed above. **(4) TermPG:** Described in Algorithm 2. We additionally implemented two variants of TermPG, including: **(5) TermPG with Reward Shaping:** We penalize the reward with a constant value upon termination. **(6) TermPG with Cost Penalty:** We penalize the reward at every time step by the optimistic cost estimator, i.e., $r - \alpha C_{\text{optimistic}}$ for some $\alpha > 0$. All TermPG variants used an ensemble of three cost networks, and a dynamic cost-dependent discount factor, as described in Section 4.2. We report mean and std. of the total reward (without penalties) for all our experiments.

**Backseat Driver (BDr).** We simulated a driving application, using MLAgents [Juliani et al., 2018], by developing a new driving benchmark, "Backseat Driver" (depicted in Figure 4), where we tested both synthetic and human terminations. The game consists of a five lane never-ending road, with randomly instantiating vehicles and coins. The agent can switch lanes and is rewarded for overtaking vehicles. In our experiments, states were represented as top view images containing the position of the agent, nearby cars, and coins with four stacked frames. We used a finite window of length 120 for termination (30 agent decision steps), mimicking a passenger forgetting mistakes of the past.

**BDr Experiment 1: Coin Avoidance.** In the first experiment of Backseat Driver, coins are considered as objects the driver must avoid. The coins signify unknown preferences of the passenger, which are not explicitly provided to the agent. As the agent collects coins, a penalty is accumulated, and the agent is terminated probabilistically according to the logistic cost model in Section 2. We emphasize

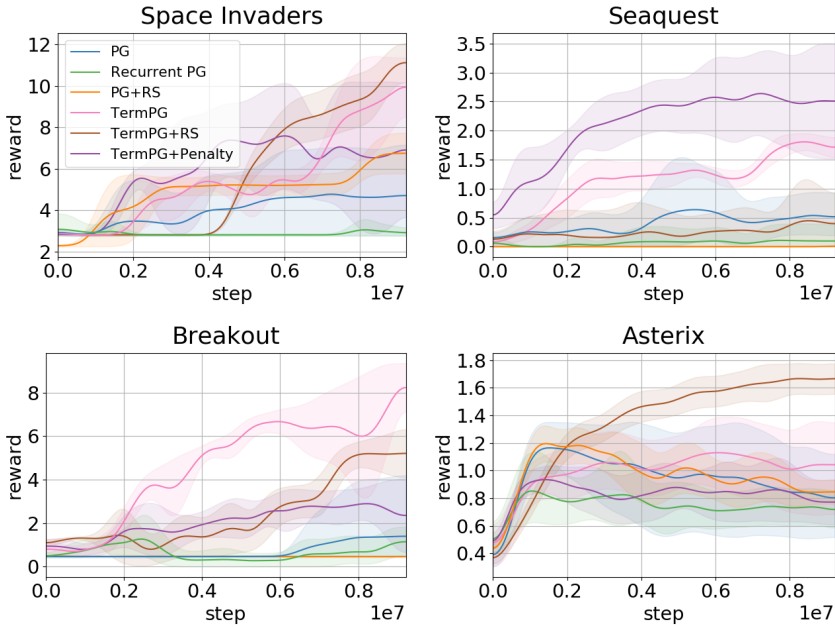

Figure 5: Results for MinAtar benchmarks. All runs were averaged over five seeds. Comparison of best performing TermPG variant to best performing PG variant (relative improvement percentage): 80% in Space Invaders, 150% in Seaquest, 410% in Breakout, and 90% in Asterix.

that, while the coins are visible to the agent (i.e., part of the agent's state), the agent only receives feedback from collecting coins through implicit terminations.

Results for Backseat Driver with coin-avoidance termination are depicted in Figure 3. We compared TermPG (pink) and its two variants (brown, purple) to the PG (blue), recurrent PG (green), and reward shaping (orange) methods described above. Our results demonstrate that TermPG significantly outperforms the history-based and penalty-based baselines. We found TermPG (pink) to perform significantly better, doubling the reward of the best PG variant. All TermPG variants converged quickly to a good solution, suggesting fast convergence of the costs (see Appendix F).

**BDr Experiment 2: Human Termination.** To complement our results, we evaluated human termination on Backseat Driver. For this, we generated data of termination sequences from agents of varying quality (ranging from random to expert performance). We asked five human supervisors to label subsequences of this data by terminating the agent in situations of "continual discomfort". This guideline was kept ambiguous to allow for diverse termination signals. The final dataset consisted of 512 termination examples. We then trained a model to predict human termination and implemented it into Backseat Driver to simulate termination. We refer the reader to

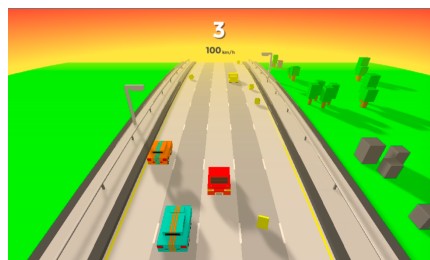

Figure 4: Backseat Driver

Appendix E for specific implementation details. Figure 3 shows results for human termination in Backseat Driver. As before, a significant performance increase was evident in our experiments. Additionally, we found that using a cost penalty (purple) or termination penalty (brown) for TermPG did not greatly affect performance.

**MinAtar.** We further compared our method to the PG, recurrent PG, and reward shaping methods, on MinAtar [Young and Tian, 2019]. For each environment, we defined cost functions that do not necessarily align with the pre-specified reward, to mimic uncanny behavior that humans are expected to dislike. For example, in Breakout, the agent was penalized whenever the paddle remained in specific regions (e.g., sides of the screen), whereas in Space Invaders, the agent was penalized for

Table 1: Summary of results (top) and ablations for TermPG (bottom). Standard deviation optimism did not have significant impact on performance. Removing optimism or the dynamic discount factor had negative impact on performance. TermPG was found to be robust to model misspecfcations of the accumulated cost window.

| | Backseat Driver | | MinAtar | | | |
|---|---|---|---|---|---|---|
| **Experiment** | **Coin Avoid.** | **Human** | **Space Inv.** | **Seaquest** | **Breakout** | **Asterix** |
| PG | $5.3 \pm 0.8$ | $4.9 \pm 1.5$ | $5.2 \pm 1.8$ | $0.6 \pm 0.4$ | $1.4 \pm 2.8$ | $0.8 \pm 0.3$ |
| Recurrent PG | $3.4 \pm 0.21$ | $5 \pm 1.8$ | $2.8 \pm 0.05$ | $0.1 \pm 0.3$ | $0.7 \pm 0.6$ | $0.7 \pm 0.2$ |
| PG + RS | $5.9 \pm 1.4$ | $7.4 \pm 1.7$ | $7.6 \pm 2.3$ | $0.4 \pm 0.2$ | $0.5 \pm 0.03$ | $0.9 \pm 0.2$ |
| TermPG (ours) | $\mathbf{8.7 \pm 1.4}$ | $8.3 \pm 1.3$ | $9.7 \pm 1.1$ | $\mathbf{1.4 \pm 0.8}$ | $\mathbf{8.2 \pm 0.3}$ | $1 \pm 0.2$ |
| TermPG + RS (ours) | $\mathbf{8.4 \pm 1.3}$ | $7.7 \pm 0.3$ | $\mathbf{11.8 \pm 0.8}$ | $0.3 \pm 0.6$ | $5.1 \pm 1$ | $0.8 \pm 0.1$ |
| TermPG + Penalty (ours) | $6 \pm 0.8$ | $\mathbf{11.8 \pm 1.5}$ | $7.7 \pm 1.4$ | $\mathbf{2.4 \pm 1}$ | $2.3 \pm 2.3$ | $\mathbf{1.7 \pm 0.1}$ |
| **Ablation Test** | **Coin Avoid.** | **Human** | **Space Inv.** | **Seaquest** | **Breakout** | **Asterix** |
| Optimism with Ensemble Std. | $7.6 \pm 2.1$ | $7.5 \pm 1.1$ | $2.8 \pm 0.02$ | $0.9 \pm 0.6$ | $10.9 \pm 1$ | $1 \pm 0.1$ |
| No Optimism | $7.8 \pm 1.3$ | $8.8 \pm 1.2$ | $5.2 \pm 1.6$ | $0.7 \pm 0.3$ | $1.3 \pm 0.7$ | $1 \pm 0.1$ |
| No Dynamic Discount | $6.9 \pm 0.6$ | $5.9 \pm 0.7$ | $4.4 \pm 1.8$ | $0.4 \pm 0.1$ | $0.5 \pm 0.02$ | $0.8 \pm 0.1$ |
| $\times 0.5$ Window Misspecification | $7.2 \pm 1.1$ | $7.2 \pm 0.5$ | $9.7 \pm 3.1$ | $2.4 \pm 0.4$ | $7.9 \pm 0.8$ | $0.8 \pm 0.1$ |
| $\times 2$ Window Misspecification | $8.3 \pm 0.1$ | $8.4 \pm 0.2$ | $11.1 \pm 3$ | $2.2 \pm 0.2$ | $10.3 \pm 0.8$ | $1 \pm 0.1$ |

"near misses" of enemy bullets. We refer the reader to Appendix E for specific details of the different termination cost functions.

Figure 5 depicts results on MinAtar. As with Backseat Driver, TermPG lead to significant improvement, often achieving a magnitude order as much reward as Recurrent PG. We found that adding a termination penalty and cost penalty produced mixed results, with them being sometimes useful (e.g., Space Invaders, Sequest, Asterix), yet other times harmful to performance (e.g., Breakout). Therefore, we propose to fine-tune these penalties in Algorithm 2. Finally, we note that training TermPG was, on average, 67% slower than PG, on the same machine. Nevertheless, though TermPG was somewhat more computationally expensive, it showed a significant increase in overall performance. A summary of all of our results is presented in Table 1 (top).

**Ablation Studies.** We present various ablations for TermPG in Table 1 (bottom). First, we tested the effect replacing the type of cost optimism in TermPG. In Section 4, cost optimism was defined using the minimum of the cost ensemble, i.e., $\min\{c_{\phi_m}\}$. Instead, we replaced the cost optimism to $C_{\text{optimistic}} = \text{mean}\{c_{\phi_m}\} - \alpha\text{std}\{c_{\phi_m}\}$, testing different values of $\alpha$. Surprisingly, this change mostly decreased performance, except for Breakout, where it performed significantly better. Other ablations included removing optimism altogether (i.e., only using the mean of the ensemble), and removing the dynamic discount factor. In both cases we found a significant decrease in performance, suggesting that both elements are essential for TermPG to work properly and utilize the estimator of the unknown costs. Finally, we tested misspecifications of our model by learning with windows that were different from the environment's real cost accumulation window. In both cases, TermPG was suprisingly robust to window misspecification, as performance remained almost unaffected by it.

## 6  Related Work

Our setup can be linked to various fields, as listed below.

**Constrained MDPs.** Perhaps the most straightforward motivations for external termination stems from constraint violation [Chow et al., 2018, Efroni et al., 2020a, HasanzadeZonuzy et al., 2020], where strict or soft constraints are introduced to the agent, who must learn to satisfy them. In these setups, which are often motivated by safety [Garcıa and Fernández, 2015], the constraints are usually known. In contrast, in this work, the costs are *unknown* and only implicit termination is provided.

**Reward Design.** Engineering a good reward function is a hard task, for which frequent design choices may drastically affect performance [Oh et al., 2021]. Moreover, for tasks where humans are involved, it is rarely clear how to engineer a reward, as human preferences are not necessarily known, and humans are non-Markovian by nature [Clarke et al., 2013, Christiano et al., 2017]. Termination can thus be viewed as an efficient mechanism to elicit human input, allowing us to implicitly interpret human preferences and utility more robustly than trying to specify a reward.

**Global Feedback in RL.** Recent work considered once-per-trajectory reward feedback in RL, observing either the cumulative rewards at the end of an episode [Efroni et al., 2020b, Cohen et al., 2021] or a logistic function of trajectory-based features Chatterji et al. [2021]. While these works are

based on a similar solution mechanism, our work concentrates on a new framework, which accounts for non-Markovian termination. Additionally, we provide per-state concentration guarantees of the unknown cost function, compared to global concentration bounds in previous work [Abbasi-Yadkori et al., 2011, Zhang et al., 2016, Qi et al., 2018, Abeille et al., 2021]. Using our local guarantees, we are able to construct a scalable policy gradient solution, with significant improvement over recurrent and reward shaping based approaches.

**Preference-based RL.** In contrast to traditional reinforcement learning, preference-based reinforcement learning (PbRL) relies on subjective opinions rather than numerical rewards. In PbRL, preferences are captured through probabilistic rankings of trajectories [Wirth et al., 2016, 2017, Xu et al., 2020]. Similar to our work, Christiano et al. [2017] use a regression model to learn a reward function that could account for the preference feedback. Our work considers a different setting in which human feedback is provided through termination, where termination and reward may not align.

# 7 Discussion

This paper formulated a new model to account for history-dependent exogenous termination in reinforcement learning. We defined the TerMDP framework and proposed a theoretically-guaranteed solution, as well as a practical policy-gradient approach. Our results showed significant improvement of our approach over various baselines. We stress that while it may seem as if the agent has two potentially conflicting goals—avoiding termination and maximizing reward—they are, in fact, aligned. The long-term consequences of actions need to account for longer survival which, in turn, allows for more reward collection. In what follows, we discuss $\kappa$, as factored in our regret bounds, as well possible limitations of our work.

**The Role of $\kappa$** As shown in Theorem 2, $\kappa$ plays a significant role in the regret bound of Algorithm 1. This linear dependence is induced from the confidence bounds of Theorem 1. Informally, $\kappa$ is negligible whenever the costs $c$ and bias $b$ are "well behaved". Suppose $\sum_{t=1}^{h} c_t(s_t^k, a_t^k) - b \gg 0$. In this case, $\kappa$ would be large and termination would mostly occur after the first step. As such, estimation of the costs would be hard (see Chatterji et al. [2021]). Alternatively, suppose $\sum_{t=1}^{h} c_t(s_t^k, a_t^k) - b \ll 0$. In this case, $\kappa$ would also be large. Here, credit assignment would make the cost estimation problem harder, as trajectories would span longer horizons. It is unclear, as to the writing of this work, if other solutions to TerMDPs could bring about stronger regret guarantees that significantly reduce their dependence on $\kappa$. We note that lower bounds, which include $\kappa$, have previously been established for the estimation problem (see Abeille et al. [2021], Faury et al. [2020], Jun et al. [2021]). Nevertheless, when searching for a policy which maximizes reward, it is unclear if estimation of the costs is indeed necessary for every state. We leave this direction for future work.

**Limitations and Negative Societal Impact** A primary limitation of our work involves the linear dependence of the logistic termination model. In some settings, it might be hard to capture true human preferences and behaviors using a linear model. Nevertheless, when measured across the full trajectory, our empirical findings show that this model is highly expressive, as we demonstrated on real human termination data (Section 5). Additionally, we note that work in inverse RL [Arora and Doshi, 2021] also assumes such linear dependence of human decisions w.r.t. reward. Future work can consider more involved hypothesis classes, building upon our work to identify the optimal tradeoff between expressivity and convergence rate.

Finally, we note a possible negative societal impact of our work. Termination is strongly motivated by humans interacting with the agent. This may be harmful if not carefully controlled, as learning incorrect or biased preferences may, in turn, result in unfavorable consequences, or if humans engage in adversarial behavior in order to mislead an agent. Our work discusses initial research in this domain. We encourage caution in real-world applications, carefully considering the possible effects of model errors, particularly in applications that affect humans.

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
