# Reinforcement Learning with a Terminator

**Guy Tennenholtz** [*][†]    **Nadav Merlis** [*]    **Lior Shani** [*]    **Shie Mannor** [*][†]
guytenn@gmail.com

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

$$h = 1 \quad \left\{ s_1^{(m)}, a_1^{(m)} \right\}_{m=1}^M \rightarrow \boxed{\text{Model Ensemble}} \rightarrow \left\{ \hat{c}_{\phi_m} \left( s_1^{(m)}, a_1^{(m)} \right) \right\}_{m=1}^M$$

$$h = 2 \quad \left\{ s_2^{(m)}, a_2^{(m)} \right\}_{m=1}^M \rightarrow \boxed{\text{Model Ensemble}} \rightarrow \left\{ \hat{c}_{\phi_m} \left( s_2^{(m)}, a_2^{(m)} \right) \right\}_{m=1}^M$$

$$\vdots$$

$$h = t \quad \left\{ s_t^{(m)}, a_t^{(m)} \right\}_{m=1}^M \rightarrow \boxed{\text{Model Ensemble}} \rightarrow \left\{ \hat{c}_{\phi_m} \left( s_t^{(m)}, a_t^{(m)} \right) \right\}_{m=1}^M$$

$\Sigma \rightarrow \rho \rightarrow$ Cross Entropy Loss

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

# Table of Contents

## ORGANIZATION OF THE APPENDIX

This appendix is organized as follows. We begin by further discussing motivations of our setting in autonomous driving and recommender system tasks in Appendix A. The first part of the appendix is then mostly focused on the empirical aspects, while the second part is mostly focused on the theoretical aspects, as well as missing proofs.

In Appendix B we discuss the TerMDP model in which costs are known. In this setting, we show that the costs are indeed sufficient for finding an optimal policy. That is, one need not care about the full history to account for termination, and only the current state and accumulated costs are needed to identify an optimal policy (i.e., achieve the same value as the optimal history-dependent policy).

In Appendix C we discuss the dynamic discount factor and derive the corresponding Termination Bellman Equations, as presented in Section 4.2. In Appendix D we discuss implementation details, including descriptions of the different environment and cost functions, TermPG implementations details and hyperparameters, and the construction of the human termination data. In Appendix E we provide some additional experimental results on the cost error and adding a cost bonus to the reward.

Appendix G is focused on the problem of approximate planning in known TerMDPs. Here, we show that, by discretizing the costs on a grid, we can achieve near optimal performance through an approximate optimistic planner.

Finally, Appendices H to K provide full proofs of Theorems 1 and 2. In Appendix H we define failure events and bonuses for optimism in Algorithm 1. Appendix I analyzes and proves the regret guarantees in Theorem 2, and Appendix J provides proof for Theorem 1. Finally in Appendix K we state auxiliary lemmas used throughout our analysis.

# A  Motivation

We begin by describing two concrete examples in which non-Markovian termination naturally occurs: overrides in autonomous driving and users abandoning recommender systems.

**Autonomous Driving.** A myriad of factors affect the quality of a driving policy. These include safety, navigation efficiency, comfort, legislation, as well as specific preferences of the passengers. As these may be difficult to characterize quantitatively, heuristic metrics are derived and optimized. When a policy is released, these unknown factors may be enforced by a driving passenger, overriding the policy. For instance, a passenger might perceive a safe driving as dangerous due to differences between human and autonomous vehicle capabilities, thus halting the driving policy. This complication may be resolved by learning how to overcome such termination from previous occurrences.

**Recommender Systems.** Companies that recommend items to users may find contrasting goals between maximizing revenue and overall user satisfaction. For example, top-selling items which are recommended repeatedly can antagonize the user. Other popular items might offend the user or undermine the recommendation engine's credibility. Similar to the autonomous driving problem, user abandonment constitutes a critical signal, which may be caused by continual user dissatisfaction. The original criteria should therefore be optimized while also learning and accounting for these non-Markovian and unknown preferences.

# B  TerMDPs with Known Costs

In this section we define the TerMDP model with known costs. We show that the accumulated costs at time step $h$, i.e., $C_h = \sum_{t=1}^{h-1} c(s_t, a_t)$ and the current state $s_h$ are a sufficient statistic for computing an optimal policy $\pi^*$. That is, the history $\tau_{1:h} = (s_1, a_1, s_2, a_2, \ldots, s_{h-1}, a_{h-1})$ can be replaced by the accumulated costs $C_h$. To see this, we define an equivalent MDP, for which the state space is augmented by the accumulated costs, and show that an optimal policy of the augmented MDP achieves the optimal value of a history dependent optimal policy of the TerMDP.

We define the augmented MDP $\mathcal{M}_{\mathrm{aug}} = (\mathcal{S}_{\mathrm{aug}}, \mathcal{A}_{\mathrm{aug}}, P_{\mathrm{aug}}, R_{\mathrm{aug}}, H)$, where $\mathcal{S}_{\mathrm{aug}} = \mathcal{S} \times \mathbb{R}$ is the augmented state space, and $\mathcal{A}_{\mathrm{aug}} = \mathcal{A}$ is the (unchanged) action space. The augmented transition function is defined for $s, C \in \mathcal{S} \times \mathbb{R}, a \in \mathcal{A}, s', C' \in \mathcal{S} \times \mathbb{R}$

$$
P_{\mathrm{aug}}(s', C' | s, C, a) = \mathbb{1}\{C' = C + c(s, a)\} \cdot \begin{cases} 1 & , s' = s = s_{\mathrm{term}} \\ \rho(C) & , s \neq s' = s_{\mathrm{term}} \\ (1 - \rho(C))P(s' | s, a) & , s, s' \neq s_{\mathrm{term}} \\ 0 & , \mathrm{o.w.} \end{cases}.
$$

Finally, the augmented reward function $R_{\mathrm{aug}}$ satisfies $r_{\mathrm{aug},h}(s_{\mathrm{aug},h}^k = (s_h^k, C), \tilde{a}_h^k) = r_h(s_h^k, a_h^k)$.

Next, consider the TerMDP with known costs $\mathcal{M}_T = (\mathcal{S}, \mathcal{A}, P, R, H, c)$, and let $C_h = \sum_{t=1}^{h-1} c(s_t, a_t)$. Then,

$$
P(s_{h+1}, C_{h+1} | s_h, a_h, \tau_{1:h}) = P(s_{h+1}, C_{h+1} | s_h, a_h, \tau_{1:h}, C_h)
$$

$$
= \mathbb{1}\{C_{h+1} = C_h + c(s_h, a_h)\} \cdot \begin{cases} 1 & , s_{h+1} = s_h = s_{\mathrm{term}} \\ \rho(C_h) & , s_h \neq s_{h+1} = s_{\mathrm{term}} \\ (1 - \rho(C_h))P(s_{h+1} | s_h, a_h) & , s_h, s_{h+1} \neq s_{\mathrm{term}} \\ 0 & , \mathrm{o.w.} \end{cases}
$$

$$
= P_{\mathrm{aug}}(s_{h+1}, C_{h+1} | s_h, C_h, a_h) = P_{\mathrm{aug}}(s_{\mathrm{aug},h+1} | s_{\mathrm{aug},h}, a_h).
$$

To prove that the costs are sufficient for optimality, we prove a more general result. Particularly, define an MDP $(\mathcal{S}_1 \times \mathcal{S}_2, \mathcal{A}, P, r, H)$, and let $f : \mathcal{S}_2 \mapsto D$, where $D$ is some known domain. Define the following set of deterministic policies

$$
\Pi_{\mathrm{aug}} = \{\pi : \mathcal{S}_1 \times \mathcal{S}_2 \mapsto \mathcal{A} : \exists \eta : \mathcal{S}_1 \times D \mapsto [0, 1], \pi(s_1, s_2) = \eta(s_1, f(s_2))\}.
$$

Define the augmented optimal value for some $s \in \mathcal{S}_1 \times \mathcal{S}_2$

$$
V_{\mathrm{aug},1}^*(s_1, s_2) = \max_{\pi \in \Pi_{\mathrm{aug}}} \mathbb{E}\left[\sum_{t=1}^H r_t(s_t, a_t) \ \middle| \ s_1 = s_1, s_2 = s_2, a_t \sim \pi_t(s_1, s_2)\right].
$$

We will show that if the reward does not depend on $\mathcal{S}_2$, and if $P(s_1', f(s_2') | s_1, s_2, a) = P(s_1', f(s_2') | s_1, f(s_2), a)$, then $V_{\mathrm{aug},1}^*(s_1, s_2) = V_1^*(s_1, s_2)$. This will prove that costs are indeed sufficient, as playing any policy in $\Pi_{\mathrm{aug}}$ in the original MDP and achieve the same value. To see this, choose $\mathcal{S}_2$ to be the set of possible trajectories in the known TerMDP, and let

$$
f(\tau_{1:h}) = f(s_1, a_1, s_2, a_2, \ldots s_{h-1}, a_{h-1}) = \sum_{t=1}^{h-1} c(s_t, a_t).
$$

Then, since we previously showed that $r_{\mathrm{aug},h}(s_{\mathrm{aug},h}^k = (s_h^k, C), \tilde{a}_h^k) = r_h(s_h^k, a_h^k)$ and $P(s_{h+1}, C_{h+1} | s_h, a_h, \tau_{1:h}) = P_{\mathrm{aug}}(s_{\mathrm{aug},h+1} | s_{\mathrm{aug},h}, a_h)$, this concludes our claim. The formal result is stated and proved below.

**Proposition 1.** *Let $\mathcal{M} = (\mathcal{S}_1 \times \mathcal{S}_2, \mathcal{A}, P, r, H)$. Assume for any $s_1, s_2 \in \mathcal{S}_1 \times \mathcal{S}_2, a \in \mathcal{A}$, $P(s_1', f(s_2') | s_1, s_2, a) = P(s_1', f(s_2') | s_1, f(s_2), a)$ and $r(s_1, s_2, a) = g(s_1, a)$, for some deterministic function $g : \mathcal{S}_1 \times \mathcal{A} \mapsto [0, 1]$. Then, for any $s_1, s_2 \in \mathcal{S}_1 \times \mathcal{S}_2$,*

$$
V_{aug,1}^*(s_1, s_2) = V_1^*(s_1, s_2).
$$

*Proof.* We prove by induction on $h \in [H]$. For $h = H$, the result follows trivially since $V_{\text{aug},H}^*(s_1, s_2) = V_H^*(s_1, s_2) = \max_a r(s_1, s_2, a)$. Next, assume that $V_{\text{aug},k+1}^*(s_1, s_2) = V_{k+1}^*(s_1, s_2)$ for some $k \in \{1, \ldots, H-1\}$ and all $s_1, s_2 \in \mathcal{S}_1 \times \mathcal{S}_2$. Then, by the Bellman Equations,

$$V_k^*(s_1, s_2) = \max_{a \in \mathcal{A}} r(s_1, s_2, a) + \sum_{s_1', s_2' \in \mathcal{S}_1 \times \mathcal{S}_2} P(s_1', s_2'|s_1, s_2, a) V_{k+1}^*(s_1', s_2')$$

$$\overset{\text{induction step}}{=} \max_{a \in \mathcal{A}} r(s_1, s_2, a) + \sum_{s_1', s_2' \in \mathcal{S}_1 \times \mathcal{S}_2} P(s_1', s_2'|s_1, s_2, a) V_{\text{aug},k+1}^*(s_1', s_2')$$

$$= \max_{a \in \mathcal{A}} g(s_1, a) + \sum_{s_1' \in \mathcal{S}_1} \sum_{C \in D} \sum_{s_2': f(s_2')=C} P(s_1', s_2'|s_1, s_2, a) V_{\text{aug},k+1}^*(s_1', s_2')$$

Next, by Lemma 1, there exists $U_{k+1}^* : \mathcal{S}_1 \times D \mapsto \mathbb{R}$ such that $V_{\text{aug},k+1}^*(s_1, s_2) = U_{k+1}^*(s_1, f(s_2))$, for all $s_1, s_2 \in \mathcal{S}_1 \times \mathcal{S}_2$. Then,

$$\sum_{C \in D} \sum_{s_2': f(s_2')=C} P(s_1', s_2'|s_1, s_2, a) V_{\text{aug},k+1}^*(s_1', s_2') = \sum_{C \in D} \sum_{s_2': f(s_2')=C} P(s_1', s_2'|s_1, s_2, a) U_{k+1}^*(s_1', f(s_2'))$$

$$= \sum_{C \in D} U_{k+1}^*(s_1', C) \sum_{s_2': f(s_2')=C} P(s_1', s_2'|s_1, s_2, a)$$

$$= \sum_{C \in D} U_{k+1}^*(s_1', C) P(s_1', f(s_2') = C|s_1, s_2, a).$$

Using the assumption, we have that $P(s_1', f(s_2') = C|s_1, s_2, a) = P(s_1', f(s_2') = C|s_1, f(s_2), a)$. Then,

$$\sum_{C \in D} \sum_{s_2': f(s_2')=C} P(s_1', s_2'|s_1, s_2, a) V_{\text{aug},k+1}^*(s_1', s_2') = \sum_{C \in D} U_{k+1}^*(s_1', C) P(s_1', f(s_2') = C|s_1, f(s_2), a)$$

$$= \sum_{s_2' \in \mathcal{S}_2} P(s_1', s_2'|s_1, f(s_2), a) V_{\text{aug},k+1}^*(s_1', s_2').$$

where the last step follows a similar argument as above. Combining the above we get that

$$V_k^*(s_1, s_2) = V_{\text{aug},k+1}^*(s_1', s_2')$$

$$= \max_{a \in \mathcal{A}} g(s_1, a) + \sum_{s_1', s_2' \in \mathcal{S}_1 \times \mathcal{S}_2} P(s_1', s_2'|s_1, f(s_2), a) V_{\text{aug},k+1}^*(s_1', s_2')$$

$$= \max_{\eta: \mathcal{S}_1 \times D \mapsto [0,1]} g(s_1, \eta(s_1, f(s_2))) + \sum_{s_1', s_2' \in \mathcal{S}_1 \times \mathcal{S}_2} P(s_1', s_2'|s_1, f(s_2), \eta(s_1, f(s_2))) V_{\text{aug},k+1}^*(s_1', s_2')$$

$$= \max_{\eta: \mathcal{S}_1 \times D \mapsto [0,1]} r(s_1, s_2, \eta(s_1, f(s_2))) + \sum_{s_1', s_2' \in \mathcal{S}_1 \times \mathcal{S}_2} P(s_1', s_2'|s_1, s_2, \eta(s_1, f(s_2))) V_{\text{aug},k+1}^*(s_1', s_2')$$

$$= \max_{\pi \in \Pi_{\text{aug}}} r(s_1, s_2, \pi(s_1, s_2)) + \sum_{s_1', s_2' \in \mathcal{S}_1 \times \mathcal{S}_2} P(s_1', s_2'|s_1, s_2, \pi(s_1, s_2)) V_{\text{aug},k+1}^*(s_1', s_2')$$

$$= V_{\text{aug},k}^*(s_1, s_2)$$

This completes the proof. $\qquad\square$

**Lemma 1.** *Let $\mathcal{M} = (\mathcal{S}_1 \times \mathcal{S}_2, \mathcal{A}, P, r, H)$. Assume for any $s_1, s_2 \in \mathcal{S}_1 \times \mathcal{S}_2$, $a \in \mathcal{A}$, $P(s_1', f(s_2')|s_1, s_2, a) = P(s_1', f(s_2')|s_1, f(s_2), a)$ and $r(s_1, s_2, a) = g(s_1, a)$, for some deterministic function $g : \mathcal{S}_1 \times \mathcal{A} \mapsto [0, 1]$. Then, for any $k \in [H]$, there exists $U_k^* : \mathcal{S}_1 \times D \mapsto \mathbb{R}$ such that $V_{\text{aug},k}^*(s_1, s_2) = U_k^*(s_1, f(s_2))$, for all $s_1, s_2 \in \mathcal{S}_1 \times \mathcal{S}_2$.*

*Proof.* We prove by induction on $k$. For $k = H$, the result follows trivially as $V_{\text{aug},H}^*(s_1, s_2) = \max_a g(s_1, a)$ for all $s_1, s_2 \in \mathcal{S}_1 \times \mathcal{S}_2$. Otherwise, we the $V_{\text{aug},k+1}^*(s_1, s_2) = U^*(s_1, f(s_2))$ is true

for some $k \in \{1, \ldots, H-1\}$. Then,

$$V_{\mathrm{aug},k}^*(s_1, s_2) = \max_{\pi \in \Pi_{\mathrm{aug}}} r(s_1, s_2, \pi(s_1, f(s_2))) + \sum_{s_1', s_2' \in \mathcal{S}_1 \times \mathcal{S}_2} P(s_1', s_2'|s_1, s_2, \pi(s_1, f(s_2))) V_{k+1}^*(s_1', s_2')$$

$$\overset{\text{induction step}}{=} \max_{\pi \in \Pi_{\mathrm{aug}}} r(s_1, s_2, \pi(s_1, f(s_2))) + \sum_{s_1', s_2' \in \mathcal{S}_1 \times \mathcal{S}_2} P(s_1', s_2'|s_1, s_2, \pi(s_1, f(s_2))) U_{k+1}^*(s_1', f(s_2'))$$

$$= \max_{\pi \in \Pi_{\mathrm{aug}}} g(s_1, \pi(s_1, f(s_2))) + \sum_{s_1' \in \mathcal{S}_1} \sum_{C \in D} \sum_{s_2': f(s_2')=C} P(s_1', s_2'|s_1, s_2, \pi(s_1, f(s_2))) U_{k+1}^*(s_1', f(s_2'))$$

$$= \max_{\pi \in \Pi_{\mathrm{aug}}} g(s_1, \pi(s_1, f(s_2))) + \sum_{s_1' \in \mathcal{S}_1} \sum_{C \in D} U_{k+1}^*(s_1', C) \sum_{s_2': f(s_2')=C} P(s_1', s_2'|s_1, s_2, \pi(s_1, f(s_2)))$$

$$= \max_{\pi \in \Pi_{\mathrm{aug}}} g(s_1, \pi(s_1, f(s_2))) + \sum_{s_1' \in \mathcal{S}_1} \sum_{f(s_2') \in D} P(s_1', f(s_2')|s_1, s_2, \pi(s_1, f(s_2))) U_{k+1}^*(s_1', f(s_2'))$$

$$= \max_{\pi \in \Pi_{\mathrm{aug}}} g(s_1, \pi(s_1, f(s_2))) + \sum_{s_1' \in \mathcal{S}_1} \sum_{f(s_2') \in D} P(s_1', f(s_2')|s_1, f(s_2), \pi(s_1, f(s_2))) U_{k+1}^*(s_1', f(s_2'))$$

By defining

$$U_k^*(s_1, f(s_2)) = \max_{\pi \in \Pi_{\mathrm{aug}}} g(s_1, \pi(s_1, f(s_2))) + \sum_{s_1' \in \mathcal{S}_1} \sum_{f(s_2') \in D} P(s_1', f(s_2')|s_1, f(s_2), \pi(s_1, f(s_2))) U_{k+1}^*(s_1', f(s_2')),$$

we see that $V_{\mathrm{aug},k}^*(s_1, s_2) = U_k^*(s_1, f(s_2))$, for any $s_1, s_2 \in \mathcal{S}_1, \mathcal{S}_2$, and the proof is complete. $\quad\square$

## C  Dynamic Discount Factor

In this section we derive the Termination Bellman Equations, as defined in Section 4.2. The state action value function for a TerMDP with known costs (Appendix B) is defined for $s, C \in \mathcal{S} \times \mathbb{R}$, $a \in \mathcal{A}$, $h \in [H]$, and any policy $\pi$ by

$$Q_h^\pi(s, C, a) = \mathbb{E}_\pi \left[ \sum_{t=h}^{H} \left( \prod_{j=h+1}^{t} \gamma_j \right) r_t(s_t, a_t) \;\middle|\; s_h = s, C_h = C, a_h = a \right],$$

where we denote

$$\gamma_j = 1 - \rho(C_j - b). \tag{2}$$

Then, for any $h \in [H - 1]$ we have that

$Q_h^\pi(s, C, a)$

$$= \mathbb{E}_\pi \left[ \sum_{t=h}^{H} \left( \prod_{j=h+1}^{t} \gamma_j \right) r_t(s_t, a_t) \;\middle|\; s_h = s, C_h = C, a_h = a \right]$$

$$= r_h(s, a) + \gamma_{h+1} \mathbb{E}_\pi \left[ \sum_{t=h+1}^{H} \left( \prod_{j=h+2}^{t} \gamma_j \right) r_t(s_t, a_t) \;\middle|\; s_h = s, C_h = C, a_h = a \right]$$

$$= r_h(s, a) + \gamma_{h+1} \mathbb{E}_{s' \sim P(\cdot|s,a), a' \sim \pi(s')} \left[ \mathbb{E}_\pi \left[ \sum_{t=h+1}^{H} \left( \prod_{j=h+2}^{t} \gamma_j \right) r_t(s_t, a_t) \;\middle|\; \begin{smallmatrix} s_h = s, C_h = C, a_h = a, \\ s_{h+1} = s', C_{h+1} = C + s(s',a'), a_{h+1} = a' \end{smallmatrix} \right] \right]$$

$$= r_h(s, a) + \gamma_{h+1} \mathbb{E}_{s' \sim P(\cdot|s,a), a' \sim \pi(s')} \left[ \mathbb{E}_\pi \left[ \sum_{t=h+1}^{H} \left( \prod_{j=h+2}^{t} \gamma_j \right) r_t(s_t, a_t) \;\middle|\; s_{h+1} = s', C_{h+1} = C + s(s',a'), a_{h+1} = a' \right] \right]$$

$$= r_h(s, a) + \gamma_{h+1} \mathbb{E}_{s' \sim P(\cdot|s,a), a' \sim \pi(s')} \left[ Q_{h+1}^\pi(s', C + c(s', a'), a') \right]$$

$$= r_h(s, a) + (1 - \rho(C - b)) \mathbb{E}_{s' \sim P(\cdot|s,a), a' \sim \pi(s')} \left[ Q_{h+1}^\pi(s', C + c(s', a'), a') \right].$$

We get the Termination Bellman Equations

$$Q_h^\pi(s, C, a) = r_h(s, a) + (1 - \rho(C - b)) \mathbb{E}_{s' \sim P(\cdot|s,a), a' \sim \pi(s')} \left[ Q_{h+1}^\pi(s', C + c(s', a'), a') \right].$$

Similarly, we can define an infinite horizon setting, for which

$$Q^\pi(s, C, a) = \mathbb{E}_\pi \left[ \sum_{t=0}^{\infty} \left( \prod_{j=2}^{t} \gamma_j \right) r_t(s_t, a_t) \;\middle|\; s_1 = s, C_1 = C, a_1 = a \right],$$

which yields the infinite horizon Termination Bellman Equations

$$Q^\pi(s, C, a) = r(s, a) + (1 - \rho(C - b)) \mathbb{E}_{s' \sim P(\cdot|s,a), a' \sim \pi(s')} [Q^\pi(s', C + c(s', a'), a')]$$

Having defined the Termination Bellman Equations, we use them for value estimation. Specifically, we consider estimating the value by minimizing the TD-error, for some $s, C, a, s', a'$

$$\left| Q(s, C, a) - r(s, a) - (1 - \rho(C - b)) Q(s', C + c(s', a'), a') \right|.$$

In our implementation, we used Generalized Advantage Estimation (GAE, Schulman et al. [2015]), which uses an exponential average over estimates of the TD-error. We used the discount factor in Equation (2) for the TD estimates. That is, an exponential average over

$$\hat{A}^{(n)} = \sum_{t=0}^{\infty} \left( \prod_{j=2}^{n-1} \gamma_j \right) r_t(s_t, a_t) + \left( \prod_{j=2}^{n} \gamma_j \right) \hat{V}(s_n) - \hat{V}(s_1).$$

See Schulman et al. [2015] for specific implementation details of GAE.

# D   Implementation Details

In this section we describe in detail the implementation details of the used environments, cost functions, algorithm, and human termination procedure.

## D.1   Environments

We begin by describing the environments and cost functions used in our paper. In all environments we used a bias $b = 6$ for termination, which was unknown to the training agent. For all our environments (except human termination, see Appendix D.3), we used a window size of $w = 30$.

**Backseat Driver.**   The environment was built using Unity's MLAgents [Juliani et al., 2018]. We incorporated a reward and cost function in the environment as follows. A reward of $+0.1$ was given to the agent for any car that was onscreen and behind it (i.e., the agent was awarded for overtaking other vehicles). The environment was set to accumulate a cost of $+1$ for every coin that was collected. No negative reward was given by the environment upon death or termination.

In Backseat Driver, the agent can take one of three actions: change lane left, change lane right, or stay in place. If the agent hits another car the episode is terminated and the agent is reset. The state is represented by a stack of three binary maps consisting of the agent position, the other vehicle positions, and the coin positions (top view). Similar to previous work, we used the past four time steps (stacked) as the agent's state. That is, at time $t$, the agent receives 12 binary maps consisting of the agent position, other vehicle positions, and coin positions for steps $t, t-1, t-2, t-3$. For efficient learning, we repeated every action four times in the environment.

**MinAtar.**   We used four of the original MinAtar benchmarks [Young and Tian, 2019], enforcing termination using a predefined cost-function. For each of the environments, we used a cost function which was dependent on the agent's position w.r.t. objects and areas of the state. Specifically, we defined costs as follows.

- **Space Invaders:** The agent controls a spaceship and must shoot other alien spacecrafts, while they also shoot bullets at the agent.

  The agent was penalized ($c = 1$) for bullets passing at distance $d = 1$ from its current position. That is, near-misses of enemy bullets penalized the player. Ideally, the agent must learn to play the game while avoiding dangerous states in which one erroneous action can lose the game.

- **Seaquest:** In this environment, the agent controls a submarine. The agent must shoot enemies, avoid hitting objects (enemies, fish, or bullets), and carefully replenish its oxygen by moving to the surface.

  The agent was penalized ($c = 1$) whenever it was positioned mid-depth. That is, denoting by $D$ the maximum reachable depth, the agent was penalized whenever it was positioned at depth $D/2$. Ideally, the agent would remain close to the surface or deep in the water, avoiding unnecessary switches, to minimize termination.

- **Breakout:** In this game, the agent controls a paddle and can move it (left or right), bouncing a ball, which can hit rows of bricks. The agent is rewarded for breaking these bricks. If the paddle misses the ball, the agent loses and resets.

  The agent was penalized ($c = 1$) whenever it was positioned at the far-most left or right positions of the screen, i.e., close to the edges of the screen. Ideally, the agent would remain close to the center of the screen, where it can quickly reach the ball, and avoid getting "stuck" at the sides of the screen.

- **Asterix:** The agent can move in any of the directions: up, down, left, or right, while avoiding spawned enemies, and picking up treasure. If the player hits an enemy, the agent loses and resets.

  The agent was penalized ($c = 1$) whenever it was positioned at distance $d = 1$ from an enemy. Similar to previous environments, this cost function was used to encourage the agent to stay away from enemies. Ideally, the agent should not be too close to enemies, for which one erroneous action could lose the game.

Table 2: Hyperparameters used to train PPO agent

| Name | Value | Comments |
|---|---|---|
| Batch size | 32 | |
| Learning rate | $5 \times 10^{-5}$ | |
| Rollout size | 1024 | |
| Num epochs | 3 | How many training epochs to do after each rollout |
| Entropy coef. | 0 | |
| kl coef | 0.2 | Initial coefficient for KL divergence |
| kl target | 0.01 | Target value for KL divergence |
| GAE $\lambda$ | 1 | The GAE (lambda) parameter |
| Num workers | 8 | |

Table 3: CNN architecture hyperparameters for agent policy and cost networks

| | |
|---|---|
| Input size | $42 \times 42$ |
| Num Filters | $16, 32, 256$ |
| Stride | $4, 4, 11$ |
| Padding | $2, 2, 1$ |
| Activations | Relu |
| Post-network FC hidden | $400, 256$ |

## D.2 TermPG

TermPG (Algorithm 2) uses two key elements - a policy gradient algorithm (`ALG-PG`), and a training procedure for learning the costs and designing a dynamic cost-dependent discount factor, as described in Section 4. For the policy gradient algorithm, we used Proximal Policy Optimization (PPO, see Schulman et al. [2017]), implemented in RLlib [Liang et al., 2018]. We mostly used the default hyperparameters – full specification is given in Table 2. We used the same hyperparemeters for the TermPG variants and the standard PG variants described in Section 5, with one important exception – the PG variants used a constant discount factor $\gamma = 0.99$, whereas the TermPG variants did not use a discount factor [6], as the dynamic discount factor was used instead (Section 4.2). The agent used a convolutional neural network (CNN), with hyperparameters given Table 3.

For learning the costs, we used a model of three identical cost networks. At every iteration, rollouts that were collected in the environments were added to a finite buffer (FIFO) for training the cost networks. We labeled the rollouts according to whether they ended in termination. For training the cost networks we then sampled a (different) rollout from the buffer for each of the cost networks. We then split each of the rollouts to their respective windows ($t^* - 1$ negative labels and 1 positive label for a trajectory ending in termination). Finally, we trained them the cost networks end-to-end using the cross entropy loss (see Figure 2). We repeated this procedure for 30 steps. We used the same network for training the cost model as we used for the PPO agent, with hyperparameters given in Table 3. Hyperparameters for the TermPG algorithm are given in Table 4.

## D.3 Human Termination Data

In this subsection we describe in detail our process of creating and training the human termination data for Backseat Driver. To begin, we used the Backseat Environment without termination (i.e., we considered the standard infinite horizon MDP). We trained an agent using PPO on the infinite horizon environment (without termination), saving snapshots of the agent during training. Once training was over we used the snapshots to generate trajectories from the different agents, constructing a large dataset of over 5000 trajectories of different quality.

To obtain termination feedback, we randomly sampled trajectories from the generated dataset and played them back to a human supervisor, who was instructed to terminate the agent. The instructions

---

[6]We also used a constant discount factor of $\gamma = 0.99$ for the ablation test of removing the dynamic discount factor in TermPG (Section 5, Table 1).

Table 4: TermPG hyperparameters

| Name | Value | Comments |
|---|---|---|
| Learning rate | $1 \times 10^{-3}$ | |
| Bonus coefficient | 1 | Coefficient used for cost bonus (optimism) |
| Num Ensemble | 3 | |
| Cost-net train steps | 30 | |
| Cost-net batch size | $t^*$ | Each network receives all windows in trajectory ($t^*$ examples for every network) |
| Replay buffer size | 1000 | Number of trajectories held in buffer |
| Window size | 30 | |

for termination were to terminate the agent whenever it drives in a manner that is uncomfortable. In many scenarios the human supervisors terminated the agent when it switched lanes back and forth for no apparent reason (e.g., not for overtaking other vehicles). We used five different human supervisors for labeling random trajectories in the dataset, generating a total of 512 positive termination examples.

Once the data generation was complete, we trained a classifier for the termination problem with different window size. We found that a window size of $w = 35$ best fit the termination accuracy (on a validation set). Finally, we used the trained model as an estimated termination signal for Backseat Driver. We emphasize that, while we used the same termination window ($w = 35$) for training the TermPG agent, we ran experiments with $\times x2$, $\times 0.5$ misspecification of window size, showing that these did not hurt performance (see Section 5, Table 1).

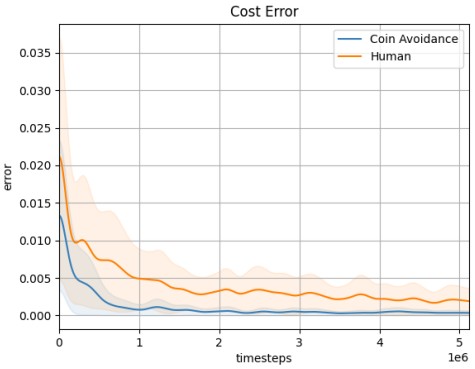

Figure 6: Cost Error for Backseat Driver

# E    Additional Results

**Cost Error.**    We plot the cost error ($l_2$ norm) of the estimated costs in Backseat Driver in both the synthetic and human termination settings in Figure 6. It can be seen that costs converge quickly, allowing for efficient learning. The speedy convergence suggests that TermPG can utilize the costs efficiently, enabling it to converge quickly to a good solution, as seen in our experiments in Section 5.

**Cost Bonus.**    In TermPG (Algorithm 2) optimism is used by augmenting the state with an optimistic cost function. Nevertheless, this information is implicit, as it only passes through the state. We tested the affect of adding optimism in costs (to encourage exploration in areas of uncertainty in costs) directly through the reward function. Specifically, we used an uncertainty (defined by the ensemble of cost networks) to add a bonus to the reward

$$r(s_t, a_t) \leftarrow r(s_t, a_t) + \alpha U(C_t),$$

where $U(C_t)$ is either the max-min difference or the standard deviation in the ensemble outputs, and $\alpha > 0$ is a hyperparameter for choosing the degree of optimism in the reward.

Results for training TermPG with a cost bonus are presented in Table 5. Evidently, adding a cost bonus mostly hurt performance, even with small values of $\alpha$. These results suggest that adding additional cost bonus to the reward is not beneficial, and using optimism in the states is sufficient.

Table 5: Cost bonus for TermPG (other TermPG results are provided for reference.)

| | Backseat Driver | | MinAtar | | | |
|---|---|---|---|---|---|---|
| **Experiment** | **Coin Avoid.** | **Human** | **Space Inv.** | **Seaquest** | **Breakout** | **Asterix** |
| TermPG | **8.7 ± 1.4** | 8.3 ± 1.3 | **9.7 ± 1.1** | **1.4 ± 0.8** | **8.2 ± 0.3** | 1 ± 0.2 |
| TermPG + RS | **8.4 ± 1.3** | 7.7 ± 0.3 | **11.8 ± 0.8** | 0.3 ± 0.6 | 5.1 ± 1 | 0.8 ± 0.1 |
| TermPG + Penalty | 6 ± 0.8 | **11.8 ± 1.5** | 7.7 ± 1.4 | **2.4 ± 1** | 2.3 ± 2.3 | **1.7 ± 0.1** |
| Cost Bonus with $\alpha = 0.1$ | 6.6 ± 0.45 | 7.9 ± 0.7 | 6.5 ± 1.46 | 1.4 ± 1.1 | 2.2 ± 0.2 | 0.9 ± 0.2 |
| Cost Bonus with $\alpha = 1$ | 7.3 ± 0.7 | 4.8 ± 1.14 | 2.8 ± 0.02 | 1.9 ± 0.1 | 0.2 ± 0.05 | 0.6 ± 0.3 |

# F   Additional Theoretical Notations

This section adds further notations needed for the sections that follow. Particularly, we define notations that are not provided in the paper for clarity, yet are beneficial for our theoretical analysis.

Throughout the paper, we work w.r.t. the natural filtration

$$\mathcal{F}_k = \sigma\Big(\big\{(s_h^1, a_h^1, R_h^1)\big\}_{h=1}^H, \ldots, \big\{(s_h^k, a_h^k, R_h^k)\big\}_{h=1}^H, s_1^{k+1}\Big).$$

We use the notation $\tau_{1:h}^k$ when referring to the trajectory of the agent at the $k^{\text{th}}$ episode. For brevity, we denote $\tau^k = \tau_{1:H}^k$. Similarly, we denote the empirical visitation up to the $h^{\text{th}}$ time step $\hat{d}_h^k(t, s, a) = \mathbb{1}\{t \le h, s_t^k = s, a_t^k = a\}$. With abuse of notation, we assume the bias term $b$ is consumed by the cost $c$, such that the first state always contains the bias term, i.e., $c(s_1, a_1) := c(s_1, a_1) - b$. Using this notation, we can write the termination probability after the $h^{\text{th}}$ time step in vector notation as $\rho(\langle \hat{d}_h^k, c \rangle)$. We then define the total transition function by $T_h^{P,c}(s'|s_h^k, a_h^k, \tau_{1:h}^k) = \Big(1 - \rho(\langle \hat{d}_h^k, c \rangle)\Big) P_h(s'|s_h^k, a_h^k)$ for any non-terminal $s' \in \mathcal{S}$ and $T_h^{P,c}(s_{\text{term}}|s_h^k, a_h^k, \tau_{1:h}^k) = \rho(\langle \hat{d}_h^k, c \rangle)$. For brevity, when working with the real kernel and costs $P, c$, we omit them from the notation and use $T_h$. Finally we denote by $\bar{X}$ optimism for some quantity $X$, e.g., $\bar{V}$ is the optimistic value.

# G Approximate Planning in TerMDPs

We consider the following planning procedure for a known TerMDP $\mathcal{M}_T = (\mathcal{S}, \mathcal{A}, P, R, H, c)$ (see Appendix B). For the approximate planning, we quantize the costs in this TerMDP to a resolution $\Delta c$. Concretely, denote the lattice of resolution $\Delta x$ by $\mathcal{G}_{\Delta x} = \{\ldots, -2\Delta x, \Delta x, 0, \Delta x, 2\Delta x, \ldots\}$. We then define the discretized cost function, for a discretization parameter $\Delta c$ as

$$c_q = \left\lfloor \frac{c}{\Delta c} \right\rfloor \Delta c$$

In this case, the accumulated cost must also lie on a grid such that $C = \sum_t c_{q,t} \in \mathcal{G}_{\Delta c}$. We denote by $c_{\max} = \max_{s,a} |c(s,a)|$. Notice that $c_{\max} \leq L$, where $L$ is known, but $c_{\max}$ is usually much smaller. We denote the number of bins of the costs by $N$, which is generally bounded by $N \leq 2\frac{c_{\max}}{\Delta c} + 1$. For non-negative costs, we will show that a much smaller $N$ suffices (Appendix G.1.2). Then, due to the grid structure, the accumulated costs can obtain at most $HN$ different values. This allows us to perform standard planning procedure in episodic MDPs (using dynamic programming/value iteration) with a state space of size $NHS$, which requires computation complexity of $\mathcal{O}(H^3 A S^2 N^2)$ [Puterman, 2014]. In the rest of this section, we show how to choose $\Delta c$ so that this approximate planner will yield a near-optimal policy. Particularly, we show that $\Delta c = \mathcal{O}(1/\sqrt{K})$ maintains the same regret guarantees (namely, incurs lower-order regret penalty) while enabling tractable planning.

## G.1 Approximation Bounds for Quantized TerMDPs

Denote the quantized TerMDP by $\mathcal{M}_T^q = (\mathcal{S}, \mathcal{A}, P, R, H, c_q)$. For a policy $\pi$, we denote by $V^\pi, V_q^\pi$ its value in $\mathcal{M}_T$ and $\mathcal{M}_T^q$, respectively. We define a policy $\pi_q$ as the solution to the planning problem in the quantized TerMDP. In what follows we show that for any $\epsilon > 0$ one can ensure $V_1^*(s_1) - V_1^{\pi_q}(s_1) < \epsilon$ for some choice of $\Delta c$.

Notice that we always rounded-down the costs (decreased the termination probability) and, thus, $V^\pi \leq V_q^\pi$ for any $\pi$. Particularly, for $\pi^*$, we get that

$$V^* = V^{\pi^*} \overset{\text{optimism}}{\leq} V_q^{\pi^*} \overset{\substack{\text{optimality of} \\ \pi_q \text{ in } \mathcal{M}_T^q}}{\leq} V_q^{\pi_q} = V^{\pi_q} + \left(V_q^{\pi_q} - V^{\pi_q}\right).$$

Therefore, it is enough to show that $V_q^{\pi_q} - V^{\pi_q} < \epsilon$. We will show that a stronger results, that $V_q^\pi - V^\pi < \epsilon$ for any $\pi$.

By the value difference lemma (Lemma 8), for any policy $\pi$

$$V_{q,1}^\pi(s_1) - V_1^\pi(s_1) = \sum_{h=1}^H \mathbb{E}\left[\left(T_h^{P,c_q} - T_h^{P,c}\right)(\cdot|s_h, a_h, \tau_{1:h})^T V_1^\pi(\cdot, \tau_{1:h+1})\right]$$

$$\leq \sum_{h=1}^H \mathbb{E}\left[\left\|\left(T_h^{P,c_q} - T_h^{P,c}\right)(\cdot|s_h, a_h, \tau_{1:h})\right\|_1 \|V_1^\pi(\cdot, \tau_{1:h+1})\|_\infty\right]$$

$$\leq H \sum_{h=1}^H \mathbb{E}\left[\left\|\left(T_h^{P,c_q} - T_h^{P,c}\right)(\cdot|s_h, a_h, \tau_{1:h})\right\|_1\right]$$

$$= H \sum_{h=1}^H \mathbb{E}\left[\left|\rho(\langle \hat{d}_h, c_q\rangle) - \rho(\langle \hat{d}_h, c\rangle)\right| \underbrace{\|P_h(\cdot|s_h, a_h)\|_1}_{=1} + \left|\left(1 - \rho(\langle \hat{d}_h, c_q\rangle)\right) - \left(1 - \rho(\langle \hat{d}_h, c\rangle)\right)\right|\right]$$

$$= 2H \sum_{h=1}^H \mathbb{E}\left[\left|\rho(\langle \hat{d}_h, c_q\rangle) - \rho(\langle \hat{d}_h, c\rangle)\right|\right]$$

### G.1.1 General Case: Signed Costs

By the Lipschitz-continuity of the logistic function, we have that

$$V_{q,1}^\pi(s_1) - V_1^\pi(s_1) \le 2H \sum_{h=1}^{H} \mathbb{E}\left[\left|\rho(\langle \hat{d}_h, c_q\rangle) - \rho(\langle \hat{d}_h, c\rangle)\right|\right]$$

$$\le \frac{H}{2} \sum_{h=1}^{H} \mathbb{E}\left[\left|\langle \hat{d}_h, c_q - c\rangle\right|\right]$$

$$\le \frac{H}{2} \sum_{h=1}^{H} \mathbb{E}\left[\underbrace{\left\|\hat{d}_h\right\|_1}_{=h} \underbrace{\|c_q - c\|_\infty}_{\le \Delta c}\right]$$

$$\le \frac{H^3 \Delta c}{2}$$

We get that, for any $\epsilon > 0$, using $\Delta c \le \frac{2\epsilon}{H^3}$ we get that $V_{q,1}^\pi(s_1) - V_1^\pi(s_1) \le \epsilon$. Note that, this choice of grid resolution induces $N = \mathcal{O}\big(H^3 \frac{c_{\max}}{\epsilon}\big)$

### G.1.2 Better Approximation for Non-Negative Costs

We remind the reader that in Appendix E we incorporated the bias $b$ into the costs $c$, such that $c(s_1, a_1) := c(s_1, a_1) - b$. This was helpful for utilizing inner product notations in our analysis. In this part, we assume the costs are positive, yet we do not limit the choice of bias $b$. Specifically, we assume that for all $h \ge 2$, and for all $s, a \in \mathcal{S} \times \mathcal{A}$, $c(s_h, a_h) \ge 0$. We further assume that for all $s, a \in \mathcal{S} \times \mathcal{A}$, $c(s_1, a_1) \ge -b$. This relates to a TerMDP with positive costs and an arbitrary bias $b$.

Let $C^* \in (0, Hc_{\max}]$ (will be explicitly chosen later). We show that, in the case of non-negative costs, it can be beneficial to clip the accumulated cost, once it is larger than $C^*$ (even if it is smaller than $Hc_{\max}$). Particularly, if $\sum_{t=1}^{h} c_{q,t}(s_t, a_t) \ge C^*$, we represent the accumulated cost by $C^*$. This, in turn, implies that the accumulated costs can have at most $N = \left\lfloor \frac{C^* + b}{\Delta c} \right\rfloor + 1$ bins.

Notice that by the definition of the logistic function, for any $C_0$ and any $C \ge C_0$, it holds that

$$\rho(C) \ge \rho(C_0) = (1 + \exp(-C_0))^{-1} \ge 1 - \exp(-C_0).$$

Therefore, since $\rho(C) \le 1$, we have for any $C \ge C_0$ that $|\rho(C) - \rho(C_0| \le \exp(-C_0)$.

Also, denote $h^* = \min\left\{h \in [H] : \langle \hat{d}_h, c_q\rangle > C^*\right\}$. Then,

$$V_{q,1}^\pi(s_1) - V_1^\pi(s_1) \le 2H \sum_{h=1}^{H} \mathbb{E}\left[\left|\rho(\langle \hat{d}_h, c_q\rangle) - \rho(\langle \hat{d}_h, c\rangle)\right|\right]$$

$$= 2H\mathbb{E}\left[\sum_{h=1}^{h^*-1} \left|\rho(\langle \hat{d}_h, c_q\rangle) - \rho(\langle \hat{d}_h, c\rangle)\right| + \sum_{h=h^*}^{H} \left|\rho(\langle \hat{d}_h, c_q\rangle) - \rho(\langle \hat{d}_h, c\rangle)\right|\right]$$

$$\le 2H\mathbb{E}\left[\frac{1}{4}(h^* - 1)^2 \Delta c + (H - h^*)\exp(-C^*)\right]$$

$$\le \frac{H^3 \Delta c}{2} + 2H^2 \exp(-C^*).$$

In the second inequality, we also used the lipschitz property of the logistic function. Moreover, we used the fact that the costs are non-negative – if the accumulated cost is larger than $C^*$ at $h^*$, then it is larger than $C^*$ for any $h \ge h^*$.

Finally, for any $\epsilon > 0$, we set $\Delta c = \frac{\epsilon}{H^3}$ and $C^* = \log\left(\frac{4H^2}{\epsilon}\right)$, which ensure an error smaller than $\epsilon$.

This parameter choice induces $N = \mathcal{O}\left(H^3 \frac{\log\left(\frac{H^2}{\epsilon}\right) + b}{\epsilon}\right)$ – potentially much smaller than the general case for which we get $\mathcal{O}\big(H^3 \frac{c_{\max}}{\epsilon}\big)$.

## G.2  Regret Bound for Approximate Planner

We integrate the approximate planner into Algorithm 1 and show that the regret bound is only mildly affected for $\epsilon = \mathcal{O}\left(1/\sqrt{K}\right)$ (or $\Delta c = \mathcal{O}\left(\frac{1}{H^3\sqrt{K}}\right)$). Particularly, instead of accurately solving the TerMDP $\left(\mathcal{S}, \mathcal{A}, H, \bar{r}^k, \hat{P}^k, \bar{c}^k\right)$, we apply our approximate planner and denote its output policy by $\pi_q^k$. Its value in the optimistic TerMDP is denoted by $\bar{V}_1^{\pi_q^k}(s_1^k)$.

Now, following the regret analysis of Appendix I, we have that

$$\text{Reg}(K) = \sum_{k=1}^{K} V_1^*(s_1^k) - V_1^{\pi_q^k}(s_1^k)$$

$$\leq \sum_{k=1}^{K} \bar{V}_1^k(s_1^k) - V_1^{\pi_q^k}(s_1^k)$$

$$= \sum_{k=1}^{K} \bar{V}_1^{\pi_q^k}(s_1^k) - V_1^{\pi_q^k}(s_1^k) + \sum_{k=1}^{K} \bar{V}_1^k(s_1^k) - \bar{V}_1^{\pi_q^k}(s_1^k)$$

The first term can be bounded exactly as in the regret analysis of Appendix I, while the second bound is controlled by the approximation error of the planner, namely $\epsilon K$. Thus, choosing $\epsilon = \mathcal{O}\left(1/\sqrt{K}\right)$ leads to a negligible second term, allowing efficient planning while maintaining the stated regret bound of Theorem 2.

# H Failure Events and Optimism

In this section we focus on defining failure events and derive the bonuses described in Section 3. Particularly, we will use these to show that the optimistic TerMDP (line 7 of Algorithm 1), used for planning, satisfies the needed optimism (i.e., larger than the optimal value), which will be used in the proof of Theorem 2.

We define the following failure events.

$$F_k^r = \left\{ \exists s, a, h : \; |r_h(s,a) - \hat{r}_h^k(s,a)| > \sqrt{\frac{2\log \frac{2SAHK}{\delta'}}{n_h^k(s,a) \vee 1}} \right\}$$

$$F_k^p = \left\{ \exists s, a, h : \; \left\| P_h(\cdot \mid s,a) - \hat{P}_h^k(\cdot \mid s,a) \right\|_1 > \sqrt{\frac{4S\log \frac{3SAHK}{\delta'}}{n_h^k(s,a) \vee 1}} \right\}$$

$$F^n = \left\{ \sum_{k=1}^K \mathbb{E}\left[ \sum_{h=1}^H \frac{1}{\sqrt{n_h^k(s_h^k, a_h^k) \vee 1}} \mid \mathcal{F}_{k-1} \right] > 16H^2 \log\left(\frac{1}{\delta'}\right) + 4SAH^2 + 2\sqrt{2}\sqrt{SAH^2 K \log HK} \right\}$$

$$F_k^c = \left\{ \exists s, a, h : \; \left| c_h(s,a) - \hat{c}_h^k(s,a) \right| > 24\sqrt{\kappa SAH^{2.5}}(L+1)^{1.5} \log^2\left( \frac{4}{\delta'}\left(1 + \frac{k(L+0.5)}{16S^2 A^2 \sqrt{H}}\right) \right) \frac{1}{\sqrt{n_h^k(s,a) + 4\frac{SAH}{L\sqrt{H}+0.5}}} \right\}$$

Furthermore, the following relations hold.

- Let $F^r = \bigcup_{k=1}^K F_k^r$. Then $\Pr\{F^r\} \le \delta'$, by Hoeffding's inequality, and using a union bound argument on all $s, a$, and all possible values of $n_k(s,a)$ and $k$. Furthermore, for $n(s,a) = 0$ the bound holds trivially since $r \in [0,1]$.

- Let $F^P = \bigcup_{k=1}^K F_k^p$. Then $\Pr\{F^p\} \le \delta'$, holds by [Weissman et al., 2003] while applying union bound on all $s, a$, and all possible values of $n_h^k(s,a)$ and $k$. Furthermore, for $n(s,a) = 0$ the bound holds trivially.

- $\Pr\{F^n\} \le \delta'$ by Lemma 9.

- Let $F^c = \bigcup_{k=1}^K F_k^c$. Then $\Pr\{F^c\} \le \delta'$ for all $k \ge 1$ by Lemma 6.

We define the good event as the event where all failure events do not occur for all $k \in [K]$, namely, $\mathcal{G} = \neg F^r \bigcap \neg F^p \bigcap \neg F^n \bigcap \neg F^c$. Then, the following holds:

**Lemma 2** (Good event). *Setting $\delta' = \frac{\delta}{4}$ then $\Pr\{F^r \bigcup F^p \bigcup F^n \bigcup F^c\} \le \delta$. When the failure events does not hold we say the algorithm is outside the failure event, or inside the good event $\mathcal{G}$.*

## H.1 Optimism

Following the events in $\mathcal{G}$ (Lemma 2) we define the following bonuses.

$$b_k^r(h,s,a) = \sqrt{\frac{2\log \frac{8SAHK}{\delta}}{n_h^k(s,a) \vee 1}}$$

$$b_k^p(h,s,a) = H\sqrt{\frac{4S\log \frac{12SAHK}{\delta}}{n_h^k(s,a) \vee 1}}$$

$$b_k^c(h,s,a) = 24\sqrt{\kappa SAH^{2.5}}(L+1)^{1.5} \log^2\left( \frac{16}{\delta}\left(1 + \frac{k(L+0.5)}{16S^2 A^2 \sqrt{H}}\right) \right) \frac{1}{\sqrt{n_h^k(s,a) + 4\frac{SAH}{L\sqrt{H}+0.5}}} \; .$$

The total reward bonus is defined as $b_k^{rp}(h,s,a) = b_k^r(h,s,a) + b_k^p(h,s,a)$. Adding the reward bonus and subtracting the cost bonus leads to an optimistic MDP, as we prove in the following lemma.

**Lemma 3** (Optimism). *Let $\bar{V}^k$ be the optimal value of the optimistic MDP $\overline{\mathcal{M}}_T(\mathcal{S},\mathcal{A},H,\bar{r}^k,\hat{P}^k,\bar{c}^k)$, clipped by $H$, i.e., for all $h \in [H], s \in \mathcal{S}$*

$$\bar{V}_h^k(s,\tau_{1:h}) = \min\left\{V_{\overline{\mathcal{M}}_T}^*(s,\tau_{1:h}), H\right\}.$$

*Then, under the good event $\mathcal{G}$, for any $k \in [K]$, $s \in [S]$, history $\tau_{1:h}$ and $h \in [H]$, it holds that $\bar{V}_h^k(s,\tau_{1:h}) \geq V^*(s,\tau_{1:h})$*

*Proof.* We prove the claim by an induction over $h$.

First, the induction trivially holds for $h = H+1$, since $\bar{V}_h^k(s,\tau_{1:h}) = V^*(s,\tau_{1:h}) = 0$.

Now, let $h \in [H]$ and assume that the claim holds for $h' = h+1$, $\forall s \in \mathcal{S}$ and $\tau_{1:h'}$.

Denote:

$$a^* \in \arg\max_a\left\{r_h(s,a) + T_h(\cdot \mid s,a,\tau_{1:h})^T V_{h+1}^*(\cdot,\tau_{1:h} \cup (s,a))\right\}.$$

Recall that $\tau_{1:h} = (s_{h-1},a_{h-1},\ldots,s_1,a_1)$ is the history up to time $h$, and for brevity, denote $\tau_{1:h+1}^* \triangleq (s,a^*) \cup \tau_{1:h} = (s,a^*,s_{h-1},a_{h-1},\ldots,s_1,a_1)$, the history up to time $h+1$ when visiting $s$ and playing $a^*$ on the $h^{th}$ timestep. Also, with some abuse of notation, we let $[TV](s,a,\tau) = T(\cdot|s,a,\tau)^T V(\cdot,(s,a)\cup\tau)$ and similarly define $[PV](s,a,\tau) = P(\cdot|s,a)^T V(\cdot,(s,a)\cup\tau)$. Finally, denote by $d^\tau$ the vector with elements $d^\tau(h,s,a) = 1$ if $(s_h,a_h) \in \tau$ and 0 otherwise. Then,

$$\bar{V}_h^k(s,\tau_{1:h}) - V_h^*(s,\tau_{1:h})$$
$$= \max_a\left\{\bar{r}_h^k(s,a) + \left[T_h^{\hat{P}_k,\bar{c}_k}\bar{V}_{h+1}^k\right](s,a,\tau_{1:h})\right\} - \max_a\left\{r_h(s,a) + \left[T_h V_{h+1}^*\right](s,a,\tau_{1:h})\right\}$$
$$\overset{(1)}{\geq} \bar{r}_h^k(s,a^*) + \left[T_h^{\hat{P}_k,\bar{c}_k}\bar{V}_{h+1}^k\right](s,a^*,\tau_{1:h}) - r_h(s,a^*) - \left[T_h V_{h+1}^*\right](s,a^*,\tau_{1:h})$$
$$\overset{(2)}{\geq} \bar{r}_h^k(s,a^*) + \left[T_h^{\hat{P}_k,\bar{c}_k}V_{h+1}^*\right](s,a^*,\tau_{1:h}) - r_h(s,a^*) - \left[T_h V_{h+1}^*\right](s,a^*,\tau_{1:h})$$
$$= \bar{r}_h^k(s,a^*) - r_h(s,a^*) + \left[\left(T_h^{\hat{P}_k,\bar{c}_k} - T_h\right)V_{h+1}^*\right](\tau_{1:h+1}^*)$$
$$= \bar{r}_h^k(s,a^*) - r_h(s,a^*) + \left[\left(T_h^{\hat{P}_k,\bar{c}_k} - T_h^{P,\bar{c}_k}\right)V_{h+1}^*\right](\tau_{1:h+1}^*) + \left[\left(T_h^{P,\bar{c}_k} - T_h\right)V_{h+1}^*\right](\tau_{1:h+1}^*)$$
$$\overset{(3)}{=} \bar{r}_h^k(s,a^*) - r_h(s,a^*)$$
$$+ \left(1 - \rho(\langle d^{\tau_{h+1}^*},\bar{c}\rangle)\right)\left[\left(\hat{P}_h - P\right)V_{h+1}^*\right](\tau_{1:h+1}^*)$$
$$+ \left(\rho(\langle d^{\tau_{h+1}^*},c\rangle) - \rho(\langle d^{\tau_{h+1}^*},\bar{c}\rangle)\right)[PV_{h+1}^*](\tau_{1:h+1}^*)$$
$$\overset{(4)}{=} \underbrace{\hat{r}_h^k(s,a^*) - r_h(s,a^*) + b_k^r(h,s,a^*)}_{(a)}$$
$$+ \underbrace{\left(1 - \rho(\langle d^{\tau_{h+1}^*},\bar{c}\rangle)\right)\left[\left(\hat{P}_h - P\right)V_{h+1}^*\right](\tau_{1:h+1}^*) + b_k^p(h,s,a^*)}_{(b)}$$
$$+ \underbrace{\left(\rho(\langle d^{\tau_{h+1}^*},c\rangle) - \rho(\langle d^{\tau_{h+1}^*},\hat{c} - b_k^c\rangle)\right)[PV_{h+1}^*](\tau_{1:h+1}^*)}_{(c)}$$

Relation $(1)$ is is due to the definition of the max-operator and the fact that $a^*$ is the optimal action in the true MDP. Next, $(2)$ is due to the induction step that $\bar{V}_h^k \geq V_h^*$. $(3)$ is by the TerMDP model assumption (and since the value at termination is 0), and $(4)$ is by the definitions of $\bar{r}_h^k(s,a)$ and $\bar{c}$.

We now turn to bound each of the three terms under the good event.

**Term (a): Reward Optimism.**

$$(a) = \hat{r}_h^k(s, a^*) - r_h(s, a^*) + b_k^r(h, s, a^*) \geq -b_k^r(h, s, a^*) + b_k^r(h, s, a^*) = 0,$$

where the first transition holds under the good event, and specifically, event $\neg F^r$).

**Term (b): Transition Optimism.**

$$
\begin{aligned}
(b) &= \left(1 - \rho(\langle d^{\tau_{h+1}^*}, \bar{c}\rangle)\right)\left[\left(\hat{P}_h - P\right)V_{h+1}^*\right](\tau_{1:h+1}^*) + b_k^p(h, s, a^*) \\
&\geq -\left|1 - \rho(\langle d^{\tau_{h+1}^*}, \bar{c}\rangle)\right|\left\|\hat{P}_h(\cdot \mid s, a^*) - P(\cdot \mid s, a^*)\right\|_1 \left\|V_{h+1}^*(\cdot, \tau_{1:h+1}^*)\right\|_\infty + b_k^p(h, s, a^*) \\
&\geq -H\left\|\hat{P}_h(\cdot \mid s, a^*) - P(\cdot \mid s, a^*)\right\|_1 + b_k^p(h, s, a^*) \\
&\geq -b_k^p(h, s, a^*) + b_k^p(h, s, a^*) \\
&= 0.
\end{aligned}
$$

The first transition is by Hölder's inequality. The second is by the fact $\forall x,\ \rho(x) \in [0, 1]$ and $\left\|V_{h+1}^*(\cdot, \tau_{1:h+1}^*)\right\|_\infty \leq H$. The third transition is by under the good event, and specifically, event $\neg F^p$.

**Term (c): Termination Cost Optimism.** First, notice that under the good event (and specifically, event $\neg F^c$, see Lemma 6) and by the definition of the bonus $b_k^c$, it holds for any $h, s, a$ that $\hat{c}_h(s, a) - b_k^c(h, s, a) \leq c_h(s, a)$. Therefore, as for any $h, s, a$ and $\tau,\ d^\tau(h, s, a) \geq 0$, it also holds that $\left\langle d^{\tau_{h+1}^*}, \hat{c} - b_k^c\right\rangle \leq \left\langle d^{\tau_{h+1}^*}, c\right\rangle$. Finally, by the monotonicity of $\rho$ and the non-negativity of $\left[PV_{h+1}^*\right](\tau_{1:h+1}^*)$, we have that

$$
\begin{aligned}
(c) &= \left(\rho(\langle d^{\tau_{h+1}^*}, c\rangle) - \rho(\langle d^{\tau_{h+1}^*}, \hat{c} - b_k^c\rangle)\right)\left[PV_{h+1}^*\right](\tau_{1:h+1}^*) \\
&\geq \left(\rho(\langle d^{\tau_{h+1}^*}, c\rangle) - \rho(\langle d^{\tau_{h+1}^*}, c\rangle)\right)\left[PV_{h+1}^*\right](\tau_{1:h+1}^*) \\
&= 0
\end{aligned}
$$

By plugging in the bounds for each of the above terms we get that

$$\bar{V}_h^k(s, \tau_{1:h}) - V_h^*(s, \tau_{1:h}) \geq 0,$$

which concludes the proof by the induction hypothesis. $\qquad\square$

# I Regret Analysis

We state the main result (Theorem 2) explicitly below. We note that in Theorem 2 we used the $\mathcal{O}$-notation, which assumes that $L = \mathcal{O}(SAH)$.

**Theorem 3** (Regret of TermCRL). *With probability at least $1 - \delta$, the regret of Algorithm 1 is*

$$\text{Reg}(K) \leq \left( 16H^2 \log\left(\frac{4}{\delta}\right) + 4SAH^2 + 2\sqrt{2}\sqrt{SAH^2 K \log HK} \right)$$

$$\times \left( 2\sqrt{2 \log \frac{8SAHK}{\delta}} + 2H\sqrt{4S \log \frac{12SAHK}{\delta}} \right.$$

$$\left. + 32H^2\sqrt{\kappa SAH}(L+1)^{1.5} \log\left( \frac{16}{\delta}\left( 1 + \frac{k(L+0.5)}{16S^2 A^2 H} \right) \right)\left( \sqrt{\frac{L+0.5}{4SAH}} \vee 1 \right) \right).$$

*Proof.* Under the good event, the regret is bounded by

$$\text{Reg}(K) = \sum_{k=1}^{K} V_1^*(s_1^k) - V_1^{\pi^k}(s_1^k)$$

$$\overset{(1)}{\leq} \sum_{k=1}^{K} \bar{V}_1^k(s_1^k) - V_1^{\pi^k}(s_1^k)$$

$$\overset{(2)}{=} \sum_{k=1}^{K} \sum_{h=1}^{H} \mathbb{E}\left[ (\bar{r}_h^k - r)(s_h^k, a_h^k) + (T_h^{\hat{P}_k, \bar{c}_k} - T_h)(\cdot | s_h^k, a_h^k, \tau_{1:h}^k)^T \bar{V}_{h+1}^{\pi^k}(\cdot, \tau_{1:h+1}^k) \,\Big|\, \mathcal{F}_{k-1} \right]$$

$$\overset{(3)}{\leq} \sum_{k=1}^{K} \sum_{h=1}^{H} \mathbb{E}\left[ (\bar{r}_h^k - r)(s_h^k, a_h^k) | \mathcal{F}_{k-1} \right] + \sum_{k=1}^{K} \sum_{h=1}^{H} \mathbb{E}\left[ \left\| (T_h^{\hat{P}_k, \bar{c}_k} - T_h)(\cdot | s_h^k, a_h^k, \tau_{1:h}^k) \right\|_1 \left\| \bar{V}_{h+1}^{\pi^k}(\cdot, \tau_{1:h+1}^k) \right\|_\infty \,\Big|\, \mathcal{F}_{k-1} \right]$$

$$\leq \underbrace{\sum_{k=1}^{K} \sum_{h=1}^{H} \mathbb{E}\left[ (\bar{r}_h^k - r)(s_h^k, a_h^k) \,\big|\, \mathcal{F}_{k-1} \right]}_{(a)}$$

$$+ \underbrace{H \sum_{k=1}^{K} \sum_{h=1}^{H} \mathbb{E}\left[ \left\| (T_h^{\hat{P}_k, \bar{c}_k} - T_h^{P, \bar{c}_k})(\cdot | s_h^k, a_h^k, \tau_{1:h}^k) \right\|_1 \,\Big|\, \mathcal{F}_{k-1} \right]}_{(b)}$$

$$+ \underbrace{H \sum_{k=1}^{K} \sum_{h=1}^{H} \mathbb{E}\left[ \left\| (T_h^{P, \bar{c}_k} - T_h)(\cdot | s_h^k, a_h^k, \tau_{1:h}^k) \right\|_1 \,\Big|\, \mathcal{F}_{k-1} \right]}_{(c)}.$$

(1) is due to the optimism of the value function (see Lemma 3) and (2) is by the value difference lemma (Lemma 8). Notice that to use this lemma, we extended the state space to include the previously visited trajectory. (3) is due to Hölder's inequality.

**Term (a): Reward Concentration.** Under the Good event (see Lemma 2), and by the definition of the bonus terms $b_k^r$ we have that

$$\sum_{k=1}^{K} \sum_{h=1}^{H} \mathbb{E}\left[ (\bar{r}_h^{k-1} - r)(s_h^k, a_h^k) \,\big|\, \mathcal{F}_{k-1} \right] \leq \sum_{k=1}^{K} \sum_{h=1}^{H} \mathbb{E}\left[ 2b_k^r(h, s_h^k, a_h^k) + b_k^p(h, s_h^k, a_h^k) \,\big|\, \mathcal{F}_{k-1} \right].$$

**Term (b): Transition Concentration.** Recall that

$$T_h^{P,c}(\cdot | s_h^k, a_h^k, \tau_{1:h}^k) = \rho\big( \langle \hat{d}_h^k, c \rangle \big) P(\cdot | s_h^k, a_h^k),$$

and since $\rho \in (0,1)$ we have that

$$\sum_{k=1}^{K}\sum_{h=1}^{H} H \mathbb{E}\left[\left\|(T_h^{\hat{P}_k, \bar{c}_k} - T_h^{P, \bar{c}_k})(\cdot | s_h^k, a_h^k, \tau_{1:h}^k)\right\|_1 | \mathcal{F}_{k-1}\right]$$

$$\leq \sum_{k=1}^{K}\sum_{h=1}^{H} H \mathbb{E}\left[\left\|(\hat{P}_k - P)(\cdot | s_h^k, a_h^k)\right\|_1 | \mathcal{F}_{k-1}\right]$$

$$\leq \sum_{k=1}^{K}\sum_{h=1}^{H} \mathbb{E}\left[b_k^p(h, s_h^k, a_h^k) | \mathcal{F}_{k-1}\right],$$

where the last transition is by the good event (Lemma 2) and the definition of $b_k^p$.

**Term (c): Termination Cost Concentration.**

$$H\sum_{k=1}^{K}\sum_{h=1}^{H}\mathbb{E}\left[\left\|(T_h^{P,\bar{c}_k} - T_h)(\cdot | s_h^k, a_h^k, \tau_{1:h}^k)\right\|_1 \,\middle|\, \mathcal{F}_{k-1}\right]$$

$$= H\sum_{k=1}^{K}\sum_{h=1}^{H}\mathbb{E}\left[\left|\rho(\langle \hat{d}_h^k, \bar{c}^k\rangle) - \rho(\langle \hat{d}_h^k, c\rangle)\right|\underbrace{\left\|P_h(\cdot | s_h^k, a_h^k)\right\|_1}_{=1} + \left|\left(1 - \rho(\langle \hat{d}_h^k, \bar{c}^k\rangle)\right) - \left(1 - \rho(\langle \hat{d}_h^k, c\rangle)\right)\right| | \mathcal{F}_{k-1}\right]$$

$$= 2H\sum_{k=1}^{K}\sum_{h=1}^{H}\mathbb{E}\left[\left|\rho(\langle \hat{d}_h^k, \bar{c}^k\rangle) - \rho(\langle \hat{d}_h^k, c\rangle)\right| \,\middle|\, \mathcal{F}_{k-1}\right]$$

$$\overset{(i)}{\leq} 2H\sum_{k=1}^{K}\sum_{h=1}^{H}\mathbb{E}\left[\left|\left\langle \hat{d}_h^k, \bar{c}^k - c\right\rangle\right| \,\middle|\, \mathcal{F}_{k-1}\right]$$

$$\overset{(ii)}{\leq} 2H\sum_{k=1}^{K}\sum_{h=1}^{H}\sum_{t=1}^{h}\mathbb{E}\left[\left|\bar{c}_t^k(s_t^k, a_t^k) - c_t(s_t^k, a_t^k)\right| \,\middle|\, \mathcal{F}_{k-1}\right]$$

$$\overset{(iii)}{\leq} 4H\sum_{k=1}^{K}\sum_{h=1}^{H}\sum_{t=1}^{h}\mathbb{E}\left[b_k^c(t, s_t^k, a_t^k) \,\middle|\, \mathcal{F}_{k-1}\right]$$

$$\leq 4H^2\sum_{k=1}^{K}\sum_{h=1}^{H}\mathbb{E}\left[b_k^c(h, s_h^k, a_h^k) \,\middle|\, \mathcal{F}_{k-1}\right],$$

$(i)$ is by the fact $\rho(\cdot)$ is 1-Lipschitz. $(ii)$ is by the fact $\forall h, s, a \; \hat{d}_h^k \in \{0, 1\}$. $(iii)$ is by the definition of $\bar{c}$ and under the good event by Lemma 6.

Combining all the above terms, we get

$$\text{Reg}(K) \leq 4\sum_{k=1}^{K}\sum_{h=1}^{H}\mathbb{E}\left[b_k^r(h, s_h^k, a_h^k) + b_k^p(h, s_h^k, a_h^k) + H^2 b_k^c(h, s_h^k, a_h^k) | \mathcal{F}_{k-1}\right].$$

Now, plugging in the bonus terms and bounding,

$$b_k^c(h, s, a) = 24\sqrt{\kappa SAH^{2.5}}(L+1)^{1.5}\log^2\left(\frac{16}{\delta}\left(1 + \frac{k(L+0.5)}{16S^2A^2\sqrt{H}}\right)\right)\frac{1}{\sqrt{n_h^k(s,a) + 4\frac{SAH}{L\sqrt{H}+0.5}}}$$

$$\leq 24\sqrt{\kappa SAH^{2.5}}(L+1)^{1.5}\log^2\left(\frac{16}{\delta}\left(1 + \frac{k(L+0.5)}{16S^2A^2\sqrt{H}}\right)\right)\frac{\left(\sqrt{\frac{L+0.5}{4SA\sqrt{H}}} \vee 1\right)}{\sqrt{n_h^k(s,a) \vee 1}},$$

we get

$$\text{Reg}(K) \le 2 \sum_{k=1}^{K} \sum_{h=1}^{H} \mathbb{E}\left[ \sqrt{\frac{2\log\frac{8SAHK}{\delta}}{n_h^k(s_k, a_k) \vee 1}} + H\sqrt{\frac{4S\log\frac{12SAHK}{\delta}}{n_h^k(s_k, a_k) \vee 1}} \,\middle|\, \mathcal{F}_{k-1} \right]$$

$$+ 4H^2 \sum_{k=1}^{K} \sum_{h=1}^{H} \mathbb{E}\left[ 24\sqrt{\kappa SAH^{2.5}}(L+1)^{1.5}\log^2\left(\frac{16}{\delta}\left(1 + \frac{k(L+0.5)}{16S^2A^2\sqrt{H}}\right)\right) \frac{\left(\sqrt{\frac{L+0.5}{4SA\sqrt{H}}} \vee 1\right)}{\sqrt{n_h^k(s,a) \vee 1}} \,\middle|\, \mathcal{F}_{k-1} \right]$$

$$\le \left( 16H^2\log\left(\frac{4}{\delta}\right) + 4SAH^2 + 2\sqrt{2}\sqrt{SAH^2K\log HK} \right) \times$$

$$\left( 2\sqrt{2\log\frac{8SAHK}{\delta}} + 2H\sqrt{4S\log\frac{12SAHK}{\delta}} \right.$$

$$\left. + 92H^2\sqrt{\kappa SAH^{2.5}}(L+1)^{1.5}\log^2\left(\frac{16}{\delta}\left(1 + \frac{k(L+0.5)}{16S^2A^2\sqrt{H}}\right)\right)\left(\sqrt{\frac{L+0.5}{4SA\sqrt{H}}} \vee 1\right) \right)$$

$$= \mathcal{O}\left( \sqrt{\kappa S^2A^2H^{8.5}L^3K\log^3\left(\frac{SAHK}{\delta}\right)} \right),$$

where the last inequality is by Lemma 9. □

## J  Cost Concentration

In the section to follow, we provide a local concentration bound for estimating the costs in the termination model. This local concentration result will allow us to use the cost bonus defined in Appendix H, which is crucial for achieving tight regret guarantees, when the costs are unknown. Specifically, this concentration result serves as the base for dealing with the cost optimism in Lemma 3 and cost concentration in Theorem 2.

We first start by describing the optimization procedure used for learning the costs (Appendix J.1). Then, by formulating this procedure as an instance of the logistic bandits problem, we obtain a concentration result for the cost, which *globally bounds the distance between the empirical and cost vectors* (Appendix J.2). Finally, we use this global concentration result together with the structure of the termination MDP to provide a local concentration bound for the costs which *holds independently for any $h, s, a$* (Appendix J.3).

Before diving into the details, we first present additional notations required to the analysis in this appendix. First, to address time steps after termination, we define $\hat{d}_h^k(t, s, a) = 0$ for any $t > t_k^*$ and $\forall t, s, a$. This allows to add these vectors to the regression problem without affecting its solution. Moreover, we denote the episode-wise Gram matrix by $A_k \triangleq \sum_{h=1}^{t^*} \hat{d}_h^k \hat{d}_h^{k^T}$ and the regularized Gram matrix up to episode $k$ by $V_k \triangleq \lambda I + \sum_{k'=1}^{k-1} A_{k'}$, for some $\lambda > 0$.

### J.1  Optimization Procedure

In a vanilla MDP, at each time step, the agent takes an action and transitions to the next state. Instead, in the termination model considered in this paper, the agent potentially gets a termination signal from the environment. Importantly, the agent can gain information about the termination probabilities even in time steps when no termination occurs. Formally, at any time step up to the time of termination $h \in [H-1]$ (the last time step always ends in termination), the agent acquires a sample from a Bernoulli random variable, where 1 means termination ($h = t_k^*$), and 0 otherwise ($h \neq t_k^*$). Since the termination probability is a logistic function of the occupancy vector, finding the termination costs can be done using a logistic regression. This leads us to solve the following optimization problem in Algorithm 1 (Line 10),

$$
\hat{c}_{k+1} \in \arg\max_{c \in \mathcal{C}} \sum_{k=1}^{K} \sum_{h=1}^{H-1} \left[ \mathbb{1}\{h < t_k^*\} \log\left(1 - \rho(\langle \hat{d}_h^k, c\rangle)\right) + \mathbb{1}\{h = t_k^*\} \log\left(\rho\left(\langle \hat{d}_h^k, c\rangle\right)\right) \right] - \lambda \|c\|_2^2,
$$

(3)

where $\mathcal{C}$ is the set of possible costs. Specifically, by the fact we only observe termination on the trajectories we played, the feedback in this problem is bandit feedback. In the next section, we use known results for the logistic bandit problem Abeille et al. [2021], and apply them to acquire a global concentration bound for the costs learned in this procedure.

### J.2  Global Cost Concentration

We start by stating concentration results for logistic bandits. Let $x_t \in \mathcal{X} \subset \mathbb{R}^d$ be a sequence of contexts s.t. $\|x\| \leq 1$ for all $x \in \mathcal{X}$ and assume that $y_t$ are Bernoulli random variable such that $\mathbb{E}[y_t|x_t] = \rho(x_t^T \theta_*)$, for some $\theta^* \in \Theta$. Next, define the regularized logistic loss w.r.t. $\lambda > 0$ by $\mathcal{L}_t(\theta) = -\sum_{s=1}^{t-1} \left(y_s \log \rho(x_s^T \theta) + (1 - y_s) \log(1 - \rho(x_s^T \theta))\right) + \lambda \|\theta\|_2^2$. We denote the unconstrained minimizer of the loss by $\bar{\theta}_t$ and its minimizer over $\Theta$ by $\hat{\theta}_t$. Finally, with slight abuse of notation, let $H_t(\theta) = \sum_{s=1}^{t-1} \dot{\mu}(x_s^T \theta) x_s x_s^T + \lambda I$.

We now focus on the confidence set $\mathcal{E}_t \delta)$, defined in [Abeille et al., 2021] as

$$
\mathcal{E}_t(\delta) = \left\{ \theta \in \Theta | \mathcal{L}_t(\theta) - \mathcal{L}_t(\bar{\theta}_t) \leq \beta_t(\delta)^2 \right\}
$$

for some appropriate $\beta_t(\delta) > 0$. By Proposition 1 and Lemma 1 of [Abeille et al., 2021], it holds that $\theta^* \in \mathcal{E}_t(\delta)$. Thus, under this event, the set is not empty. In particular, when the set is not empty, it must also contains $\hat{\theta}_t$, as the constrained minimizer of the cost. Combining this conclusion with Lemma 1 of [Abeille et al., 2021], we obtain the following lemma.

**Lemma 4** (Logistic Regression Concentration). *With probability of at least $1 - \delta$, for all $t \geq 1$,*

$$\left\|\hat{\theta}_t - \theta_*\right\|_{H_t(\theta_*)} \leq (2 + 2L)\gamma_t(\delta) + 2\sqrt{1+L}\left(\gamma_t(\delta) + \frac{\gamma_t(\delta)^2}{\sqrt{\lambda}}\right) , \tag{4}$$

*where $L = \max_{\theta \in \Theta} \|\theta\|_2$ and $\gamma_t(\delta) = \sqrt{\lambda}\left(L + \frac{1}{2}\right) + \frac{d}{\sqrt{\lambda}}\log\left(\frac{4}{\delta}\left(1 + \frac{t}{16d\lambda}\right)\right)$.*

This result can be applied to our algorithm as follows:

**Corollary 1** (Global Cost Bound). *Let $\lambda = \frac{d}{L\sqrt{H}+0.5}$, $\mathcal{X} = \left\{x \in \mathbb{R}^{SAH} : x_i \in \{0,1\}, \sum_i x_i \leq H\right\}$ and $\kappa = \left(\min_{x \in \mathcal{X}} \dot{\mu}(x^T c)\right)^{-1}$. With probability of at least $1 - \delta$, it holds that*

$$\left\|\hat{c}^k - c\right\|_{V_k} \leq 12\sqrt{\kappa SAH^{2.5}}(L+1)^{1.5}\log^2\left(\frac{4}{\delta}\left(1 + \frac{k(L+0.5)}{16S^2A^2\sqrt{H}}\right)\right).$$

*Proof.* We apply Lemma 4 with $x_{H(k-1)+h} = \hat{d}_h^k$ if $h \leq \min(t_k^*, H-1)$ and $x_{H(k-1)+h} = 0$ otherwise, where $k > 0$ is the episode and $h \in [H]$ is the step within the episode. Notice that zero contexts $x_{H(k-1)+h} = 0$ do not affect the regression solution, so steps after termination are ignored. Also, in our case, we have $\|x\|^2 \leq H$, instead of $\|x\| \leq 1$, so to compensate it, the parameter $L$ will be scaled by $\sqrt{H}$. Under our assumptions, we have $d = SAH$, and thus $\gamma_k(\delta) \leq 2\sqrt{SAH(L\sqrt{H}+0.5)}\log\left(\frac{4}{\delta}\left(1 + \frac{k(L\sqrt{H}+0.5)}{16S^2A^2H}\right)\right)$ and

$$\left\|\hat{c}^k - c\right\|_{H_t(c)} \leq (2 + 2L\sqrt{H})\gamma_t(\delta) + 2\sqrt{1 + L\sqrt{H}}\left(\gamma_t(\delta) + \frac{\gamma_t(\delta)^2}{\sqrt{\lambda}}\right)$$

$$\leq 4(1 + L\sqrt{H})\gamma_t(\delta) + \frac{\sqrt{1 + L\sqrt{H}}}{\sqrt{\lambda}}\gamma_t(\delta)^2$$

$$\leq 12\sqrt{SAH}(L\sqrt{H}+1)^{1.5}\log^2\left(\frac{4}{\delta}\left(1 + \frac{k(L\sqrt{H}+0.5)}{16S^2A^2H}\right)\right)$$

$$\leq 12\sqrt{SAH^{2.5}}(L+1)^{1.5}\log^2\left(\frac{4}{\delta}\left(1 + \frac{k(L+0.5)}{16S^2A^2\sqrt{H}}\right)\right)$$

We conclude the proof by using the fact that $\kappa H_k(c) \succeq V_t$ (Lemma 5) for $\theta^* = c$. $\qquad\square$

**Lemma 5** (Connection between Local and Global Design Matrices). *For any $\theta$, if $\kappa_{\mathcal{X}}(\theta) = \frac{1}{\min_{x \in \mathcal{X}} \dot{\mu}(x^T\theta)}$, then it holds that*

$$\kappa_{\mathcal{X}}(\theta)H_t(\theta) \succeq V_t .$$

*Proof.* For any $\theta$,

$$H_t(\theta) = \sum_{s=1}^{t-1} \dot{\mu}(x_s^T\theta)x_sx_s^T + \lambda_t I$$

$$\succeq \min_{x \in \mathcal{X}} \dot{\mu}(x_s^T\theta)\sum_{s=1}^{t-1} x_sx_s^T + \lambda_t I$$

$$= \frac{1}{\kappa_{\mathcal{X}}(\theta)}\sum_{s=1}^{t-1} x_sx_s^T + \lambda_t I$$

$$\overset{1<\kappa_{\mathcal{X}}(\theta)}{\succeq} \frac{1}{\kappa_{\mathcal{X}}(\theta)}\left(\sum_{s=1}^{t-1} x_sx_s^T + \lambda_t I\right) = \frac{1}{\kappa_{\mathcal{X}}(\theta)}V_t .$$

$\square$

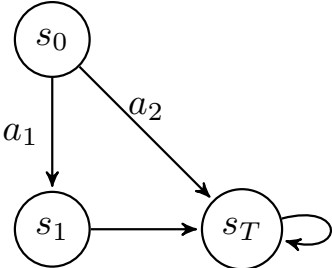

Figure 7: Example for the necessity of step-wise feedback for local estimation. Plot depicts a deterministic MDP which always begins at state $s_0$. Assume that we only receive a binary signal of whether there was termination at the end of the episode, but do not observe it during the episode. If we only observe the trajectory $s_0 \to s_1 \to s_T$, identifying the state in which the termination occurred is not possible, until we further observe the trajectory $s_0 \to s_T$. That is, there is no way to know whether termination occurred due to state $s_0$ or $s_1$. In contrast, in the TerMDP model we view termination signals at every iteration, allowing us to identify the costs locally w.r.t. every state and action.

## J.3 From Global to Local Cost Concentration

In our setting, the global cost concentration also implies a component-wise (local) concentration of the costs. This is in stark contrast to standard regression results, where specific coordinates cannot always be recovered. To see the intuition behind it, assume for simplicity that we observe the exact termination probability (rather than just a random termination signal). Notably, on each time step up to the termination, we could directly reconstruct the partial sums $\sum_{t=1}^{h} c_t(s_t^k, a_t^k)$ by inverting the logistic function. Then, the costs of all visited states could be directly calculated through the difference between the cumulative costs of any two consecutive steps. Notice that we can calculate this difference only since we get feedback on *every* time step up to termination. Were we only to observe feedback on the cumulative costs at the termination time, then we would not be able to guarantee to reconstruct anything but the global cumulative costs we were given (see example in Figure 7). In this case, the global (vector-wise) concentration used in logistic bandits (see Corollary 1 in Appendix J.2) cannot not be improved.

**Lemma 6** (Local Cost Estimation Confidence Bound). *For any $\delta > 0$, with probability of at least $1 - \delta$, for any episode $k \in [K]$, it holds that for all $h \in [H-1], s \in \mathcal{S}, a \in \mathcal{A}$,*

$$\left| \hat{c}_h^k(s,a) - c_h(s,a) \right| \leq 24\sqrt{\kappa S A H^{2.5}}(L+1)^{1.5} \log\left( \frac{4}{\delta} \left( 1 + \frac{k(L+0.5)}{16 S^2 A^2 \sqrt{H}} \right) \right) \frac{1}{\sqrt{n_h^k(s,a) + 4\frac{SAH}{L\sqrt{H}+0.5}}}.$$

*Proof.* For any $k \in [K]$, $h \in [H-1]$, $s \in \mathcal{S}$ and $h \in \mathcal{S}$, we can bound

$$\left| c_h^k(s,a) - c_h(s,a) \right| = |\langle e_{h,s,a}, \hat{c}_k - c \rangle| \leq \|e_{h,s,a}\|_{V_k^{-1}} \|\hat{c}_k - c\|_{V_k}, \tag{5}$$

where $e_{h,s,a} \in \mathbb{R}^{HSA}$ is a unit vector in the $(h,s,a)$ coordinate, and the inequality is due to Cauchy-Schwartz. We now turn to bound $\|e_{h,s,a}\|_{V_k^{-1}}$.

$$
\begin{aligned}
V_k^{-1} &\preceq \left( \lambda I + \sum_{k'=1}^{k-1} A_{k'} \mathbb{1}\left\{ (h,s,a) \in \tau^{k'} \right\} \right)^{-1} \\
&= \left( \sum_{k'=1}^{k-1} \left( \frac{\lambda}{n_h^{k-1}(s,a)} I + A_{k'} \right) \mathbb{1}\left\{ (h,s,a) \in \tau^{k'} \right\} \right)^{-1} \\
&\preceq \frac{1}{\left( n_h^{k-1}(s,a) \right)^2} \sum_{k'=1}^{k-1} \left( \frac{\lambda}{n_h^{k-1}(s,a)} I + A_{k'} \right)^{-1} \mathbb{1}\left\{ (h,s,a) \in \tau^{k'} \right\},
\end{aligned}
$$

where $n_h^{k-1}(s,a) = \sum_{k'=1}^{k-1} \mathbb{1}\left\{ (h,s,a) \in \tau^{k'} \right\}$ and the third transition is due to HM-AM inequality for positive matrices Bhagwat and Subramanian [1978].

Using this relation, we can bound $\|e_{h,s,a}\|^2_{V_k^{-1}}$ by bounding the maximal eigenvalue of each of the summands. We do so in Lemma 7, and obtain the following bound:

$$\|e_{h,s,a}\|^2_{V_k^{-1}} = e_{h,s,a}^T V_k^{-1} e_{h,s,a}$$

$$\leq \frac{1}{\left(n_h^{k-1}(s,a)\right)^2} \sum_{k'=1}^{k-1} e_{h,s,a}^T \left(\frac{\lambda}{n_h^{k-1}(s,a)} I + A_{k'}\right)^{-1} \mathbb{1}\left\{(h,s,a) \in \tau^{k'}\right\} e_{h,s,a}$$

$$\leq \frac{1}{\left(n_h^{k-1}(s,a)\right)^2} \sum_{k'=1}^{k-1} \frac{1}{\frac{1}{4} + \frac{\lambda}{n_h^{k-1}(s,a)}} \mathbb{1}\left\{(h,s,a) \in \tau^{k'}\right\} \qquad \text{(Lemma 7)}$$

$$= \frac{n_h^{k-1}(s,a)}{\left(n_h^{k-1}(s,a)\right)^2} \frac{1}{\frac{1}{4} + \frac{\lambda}{n_h^{k-1}(s,a)}}$$

$$= \frac{4}{n_h^{k-1}(s,a) + 4\lambda}.$$

By plugging into eq. (5), we obtain that for any $k$ and any $h, s, a$

$$\left|c_h^k(s,a) - c_h(s,a)\right| \leq \|\hat{c}_k - c\|_{V_k} \|e_{h,s,a}\|_{V_k^{-1}} \leq \|\hat{c}_k - c\|_{V_k} \sqrt{\frac{4}{n_h^{k-1}(s,a) + 4\lambda}}.$$

Finally, by Corollary 1 with $d = SAH$, with probability of at least $1 - \delta$ it holds that

$$\left|c_h^k(s,a) - c_h(s,a)\right| 24\sqrt{\kappa SAH^{2.5}}(L+1)^{1.5} \log\left(\frac{4}{\delta}\left(1 + \frac{k(L+0.5)}{16S^2A^2\sqrt{H}}\right)\right) \frac{1}{\sqrt{n_h^k(s,a) + 4\frac{SAH}{L\sqrt{H}+0.5}}},$$

which concludes the proof.

$\square$

**Lemma 7** (Inverse Eigenvalues Bound). *If $(h,s,a) \in \tau^k$ and $e_{h,s,a} \in \mathbb{R}^{HSA}$ is a unit vector in the coordinate $(h,s,a)$, then $e_{h,s,a}^T(\lambda I + A_k)^{-1} e_{h,s,a} \leq \frac{1}{\frac{1}{4}+\lambda}$ .*

*Proof.* Throughout the proof, we assume for brevity that there was termination in all episodes, i.e., $t_k^* \leq H - 1$ for all $k \in [K]$. Otherwise, the exact same proof follows by replacing $t_k^*$ by $\min\{t_k^*, H - 1\}$ (namely, treating the lack of termination feedback at the last timestep of the episodes). With some abuse of notations, we also use $e_i \in \mathbb{R}^{HSA}$ to denote the unit vector in the $i$-th coordinate.

To simplify the proof, it would be helpful to assume that the $t$-th coordinate of the empirical occupancy vector represents the state that was visited on the $t$-th time step (namely, coordinates are sorted by their visitation order). States that were not visited can be arbitrarily ordered. Formally, this can be done using any permutation matrix $P_k$ such that $e_{t,s_t^k,a_t^k} = P_k e_t$ for all $t \in [t_k^*]$ (and other coordinates can be arbitrarily permuted). In particular, denoting $\bar{e}_t = \sum_{i=1}^t e_i = (\underbrace{1, \ldots, 1}_{t-\text{times}}, 0, \ldots, 0)^T$, we can

write $\hat{d}_t^k = \sum_{i=1}^t e_{i,s_i^k,a_i^k} = \sum_{i=1}^t P_k e_i = P_k \bar{e}_t$.

Then, we have that

$$e_{h,s,a}^T(\lambda I + A_k)^{-1}e_{h,s,a} = e_{h,s,a}^T\left(\lambda I + \sum_{t=1}^{t_k^*}\hat{d}_t^k\hat{d}_t^{k^T}\right)^{-1}e_{h,s,a}$$

$$= e_{h,s,a}^T\left(\lambda I + \sum_{t=1}^{t_k^*}P_k\bar{e}_t\bar{e}_t^T P_k^T\right)^{-1}e_{h,s,a}$$

$$= e_{h,s,a}^T\left(P_k\left(\lambda I + \sum_{t=1}^{t_k^*}\bar{e}_t\bar{e}_t^T\right)P_k^T\right)^{-1}e_{h,s,a}$$

$$= e_{h,s,a}^T P_k\left(\lambda I + \sum_{t=1}^{t_k^*}\bar{e}_t\bar{e}_t^T\right)^{-1}P_k^T e_{h,s,a} \ ,$$

where the two last relations is since permutation matrices are orthogonal, namely $P_k^{-1} = P_k^T$.

Now, notice that $P_k$ permutes the first $t_k^*$ components to the visited $(t, s_t^k, a_t^k)$ tuples. Thus, its inverse $P_k^T$ permutes visited tuples $(t, s_t^k, a_t^k)$ to the $t$-th coordinate, namely $P_k^T e_{t,s_t^k,a_t^k} = e_t$ and

$$e_{h,s,a}^T(\lambda I + A_k)^{-1}e_{h,s,a} = e_h^T\left(\lambda I + \sum_{t=1}^{t_k^*}\bar{e}_t\bar{e}_t^T\right)^{-1}e_h \ .$$

Moreover, $\lambda I + \sum_{t=1}^{t_k^*}\bar{e}_t\bar{e}_t^T$ is a block-diagonal matrix whose first block is located at its first $t_k^*$ coordinates. Thus, each block can be inverted independently, and as $e_h$ is located in the first block of the matrix, w.l.o.g. we can only focus on the $t_k^* \times t_k^*$ first-block of the matrix. We denote this block by $B \in \mathbb{R}^{t_k^* \times t_k^*}$, and if $u_h \in \mathbb{R}^{t_k^*}$ is a unit vector in the $h^{\text{th}}$ coordinate, we can write $e_h^T\left(\lambda I + \sum_{t=1}^{t_k^*}\bar{e}_t\bar{e}_t^T\right)^{-1}e_h = u_h^T(\lambda I + B)^{-1}u_h$.

Directly calculating of the sum $\sum_{t=1}^{t_k^*}\bar{e}_t\bar{e}_t^T$, one can easily see that $B(i,j) = t_k^* + 1 - \max\{i,j\}$, as we now illustrate:

$$\begin{pmatrix} t_k^* & t_k^*-1 & t_k^*-2 & \dots & 1 \\ t_k^*-1 & t_k^*-1 & t_k^*-2 & \dots & 1 \\ t_k^*-2 & t_k^*-2 & t_k^*-2 & \dots & 1 \\ \vdots & \vdots & \vdots & \vdots & \vdots \\ 1 & 1 & 1 & 1 & 1 \end{pmatrix}$$

This matrix can be easily diagonalized, which can be used to calculate its inverse:

$$\left(\begin{array}{ccccc|ccccc} t_k^* & t_k^*-1 & t_k^*-2 & \dots & 1 & 1 & 0 & 0 & \dots & 0 \\ t_k^*-1 & t_k^*-1 & t_k^*-2 & \dots & 1 & 0 & 1 & 0 & \dots & 0 \\ t_k^*-2 & t_k^*-2 & t_k^*-2 & \dots & 1 & 0 & 0 & 1 & \dots & 0 \\ \vdots & \vdots & \vdots & \vdots & \vdots & \vdots & \vdots & \vdots & \vdots & \vdots \\ 1 & 1 & 1 & 1 & 1 & 0 & 0 & 0 & \dots & 1 \end{array}\right)$$

$$= \left(\begin{array}{ccccc|ccccc} 1 & 0 & 0 & \dots & 0 & 1 & -1 & 0 & \dots & 0 \\ 1 & 1 & 0 & \dots & 0 & 0 & 1 & -1 & \dots & 0 \\ 1 & 1 & 1 & \dots & 0 & 0 & 0 & 1 & \dots & 0 \\ \vdots & \vdots & \vdots & \vdots & \vdots & \vdots & \vdots & \vdots & \vdots & \vdots \\ 1 & 1 & 1 & 1 & 1 & 0 & 0 & 0 & \dots & 1 \end{array}\right)$$

$$= \left(\begin{array}{ccccc|ccccc} 1 & 0 & 0 & \dots & 0 & 1 & -1 & 0 & \dots & 0 \\ 0 & 1 & 0 & \dots & 0 & -1 & 2 & -1 & \dots & 0 \\ 0 & 0 & 1 & \dots & 0 & 0 & -1 & 2 & \dots & 0 \\ \vdots & \vdots & \vdots & \vdots & \vdots & \vdots & \vdots & \vdots & \vdots & \vdots \\ 0 & 0 & 0 & 0 & 1 & 0 & 0 & 0 & \dots & 2 \end{array}\right)$$

In the first step, we subtracted rows $i + 1$ from the $i$ rows, and in the second step, we subtracted the $i - 1$ rows from the $i$ rows. Thus, the inverse can be explicitly written as follows:

$$B_{i,j}^{-1} = \begin{cases} 1 & i = j = 1 \\ 2 & i = j > 1 \\ -1 & i = j - 1 \text{ or } i = j + 1 \\ 0 & \text{o.w.} \end{cases}$$

Notice that the absolute values of all rows is smaller than $4$. Then (e.g., by Gershgorin circle theorem), $\lambda_{\max}(B^{-1}) \leq 4$, and since $B$ is PSD, $\lambda_{\min}(B) \geq \frac{1}{4}$. Finally, we get the desired result by bounding

$$e_{h,s,a}^T (\lambda I + A_k)^{-1} e_{h,s,a} = u_h^T (\lambda I + B)^{-1} u_h$$

$$\leq \underbrace{\|u_h\|_2}_{=1} \lambda_{\max}\left((\lambda I + B)^{-1}\right)$$

$$= \frac{1}{\lambda_{\min}(\lambda I + B)} \leq \frac{1}{\frac{1}{4} + \lambda} \quad .$$

**Remark.** Notice that the same conclusion still holds in the extreme case of $t^* \leq 2$: we get $\lambda_{\max}(B^{-1}) \leq 3$ for $t^* = 2$ and $\lambda_{\max}(B^{-1}) = 1$ (as $B$ contains a single element). $\qquad \square$

# K    Useful Lemmas

**Lemma 8** (Value difference lemma, e.g., Dann et al. [2017], Lemma E.15). *Consider two MDPs* $\mathcal{M} = (\mathcal{S}, \mathcal{A}, P, r, H)$ *and* $\mathcal{M}' = (\mathcal{S}, \mathcal{A}, P', r', H)$. *For any policy* $\pi$ *and any* $s, h$, *the following relation holds:*

$$V_h^\pi(s; \mathcal{M}) - V_h^\pi(s; \mathcal{M}')$$
$$= \mathbb{E}\left[\sum_{h'=h}^{H} (r_{h'}(s_{h'}, a_{h'}) - r'_{h'}(s_{h'}, a_{h'})) + (P - P')(\cdot \mid s_{h'}, a_{h'})^T V_{h'+1}^\pi(\cdot; \mathcal{M}')|s_h = s, \pi, P\right]$$

The following lemma is due to Efroni et al. [2020b], with the only exception that the original lemma assumes a stationary MDP and therefore, $S$ translates to $SH$ in the following.

**Lemma 9** (Expected Cumulative Visitation Bound, Lemma 22, Efroni et al. [2020b]). *Let* $\{\mathcal{F}_k\}_{k=1}^{K}$ *be the natural filtration. Then, with probability greater than* $1 - \delta$ *it holds that*

$$\sum_{k=1}^{K} \mathbb{E}\left[\sum_{h=1}^{H} \frac{1}{\sqrt{n_h^k(s_h^k, a_h^k) \vee 1}} \mid \mathcal{F}_{k-1}\right] \leq 16H^2 \log\left(\frac{1}{\delta}\right) + 4SAH^2 + 2\sqrt{2}\sqrt{SAH^2K \log HK}$$
$$= \mathcal{O}\left(H\left(HSA + H\log\left(\frac{1}{\delta}\right)\right) + \sqrt{SAH^2K \log HK}\right)$$
$$= \tilde{\mathcal{O}}\left(\sqrt{H^2SAK}\right)$$