# OpenReview forum: "Reinforcement Learning with a Terminator"
_NeurIPS.cc/2022/Conference — NeurIPS 2022 Accept_

### Official Review · Reviewer_nYZQ · 2022-07-07

**Rating:** 7
**Confidence:** 3
**Soundness:** 4 excellent
**Presentation:** 3 good
**Contribution:** 3 good

**Summary:**

The paper introduces a variant of an episodic MDP where exogenous termination occurs - the episode is terminated by an observer based on some accumulated cost function that is not observed by the agent. Under this new formulation (and some additional assumptions) the paper proposes a solution to the problem, which amounts to estimating these unknown costs from the sparse termination signal, and then incorporating the cost estimation into an RL algorithm optimistically. Theoretical guarantees are provided with respect to the cost estimation, bounds on the regret and the existence of optimal policies in this setting. Results on several domains (requiring deep RL methods) demonstrate that the proposed algorithm outperforms policy gradient baselines when both algorithmic and human termination is applied exogenously.

**Questions:**

While reading the paper, parts of it put me in mind of other formalisms or approaches. While it is not necessary to include these in the paper, I would be curious to hear the authors' thoughts on how their paper links to those and whether there is any connection. I also have a couple of clarifying questions below.

1. The approach to learning the cost function relies on the termination condition being of a specific form. This paper introduces this MDP extension and so it is reasonable for the paper to make any claim about how termination occurs, but I am curious about other forms of termination. For example, if there was just a hard threshold at which point termination occurs, could we use something like RUDDER [1] to do credit assignment on the termination signal (1/0) to estimate the costs in practice?

2. The MDP formulation here puts me in mind of the average reward SMDP formulation of [2], where the aim is to learn a policy that maximises the infinite-horizon ratio between expected reward and expected cost. Of course, this assumes that the cost function is known, but does cater for the infinite horizon MDP case, which seems more in line with the autonomous driving example. I would be interested if the authors think there is any link here or under what conditions it would be appropriate to use the TerMDP vs AR SMDP formulation.

3. On line 59, the rewards received by the agent are in the range $[0, 1]$, but I could not see why this would be the case. Certainly, I understand that the rewards must be non-negative since termination is "bad", but why should the rewards be bounded?

4. On the topic of the theoretical bounds, I struggle to get some intuition about the various terms. The constant $L$ bounds the norm of the costs, but there is no bound on the costs themselves. So let's say that, for example, the costs were in the millions (and so was the bias). What would that mean for the bounds? Would things change if everything was scaled accordingly (e.g. costs/bias are 1 vs 1e6). A similar question about $\kappa$ - if the derivative is near 0, then $\kappa$ would be massive. Presumably, this equates to the case where termination happens extremely infrequently (or all the time)? One concern here for me would be what this means in practice - how useful are these bounds for practical purposes if termination happens every so often (ranging from all the time to never)? And how likely is termination to occur in the domains tested?

5. In the results, I notice that a window size twice as large as necessary improves performance. Is this a statistically significant result? If so, why should this be?

Minor typo: Line 158: rollouts -> rolls out

[1] Arjona-Medina, Jose A., et al. "Rudder: Return decomposition for delayed rewards." Advances in Neural Information Processing Systems 32 (2019).
[2] Ross, Sheldon M. "Average cost semi-Markov decision processes." Journal of Applied Probability 7.3 (1970): 649-656.

**Limitations:**

One big positive of the paper was that it was honest about where problematic areas arise (both in the theoretical analysis as well as potential practical deployment). One additional aspect that bears discussing, though, is [3] and related work. In particular, if termination is done from outside the system, then there is no problem. However, you could imagine a case of a real robot being operated by a real human. Since termination is bad, the robot could learn either positive behaviours that mean the human does not wish to terminate it, or it could learn to prevent the human from terminating it (e.g. by preventing the human from pressing a kill-switch), which of course would be a terrible outcome.

[3] Orseau, Laurent, and Stuart Armstrong. "Safely interruptible agents." Proceedings of the Thirty-Second Conference on Uncertainty in Artificial Intelligence. 2016.

**Strengths And Weaknesses:**


\+ The paper introduces a useful variant of the standard MDP formalism. I found this well-motivated with several real-world use cases, and I was surprised to see that no one has considered such an extension previously.

\+ The paper provides theoretical guarantees for the algorithm and then extends it to implement a practical policy gradient method that can be used on larger MDPs with all the latest deep RL algorithms.

\+ On the whole, the paper was extremely well-written (but see below) and well laid out

\+ At various points, the paper points out limitations or shortcomings, which I appreciated

\- The only section I found difficult to read was Section 3. In particular, the various symbols (such as $\kappa$) encapsulate an awful lot, and it makes it hard to get a sense of how tight the bounds are and how they behave under various conditions (I have some questions in the next section).

\- Much of the theoretical analysis relies on the particular form of termination assumed. There are also certain other assumptions made where it is not clear if they are done for simplicity or because of some fundamental reason.

\- A lesser issue, but the experiments, while requiring a neural network for function approximation, are on "smaller" domains - it is thus unclear how the method scales (or whether it can, given the computational overhead) to larger ones.

---

> ### Author Response · Authors · 2022-08-02
> **Response**
>
> Thank you for your thorough and positive review! We are encouraged that you found the problem well-motivated and the paper well written, and that you highlighted the theoretical guarantees and practical algorithm. We appreciate your comments and address them below.
>
> ### 1. Re RUDDER:
>
> We note that our problem can’t be directly cast to a delayed reward setting since the costs affect the transition function (and not the reward), but more importantly – our problem is non-Markov w.r.t. the costs.
>
> ### 2. Re relation to SMDP formulation:
>
> You raise an interesting comment w.r.t. the SMDP formulation. There is indeed a similar tradeoff between costs and rewards in that formulation as we have in our setting. On the one hand, lowering the termination probability directly improves the overall reward merely by extended survival. On the other hand, assuming for the sake of simplicity that the cost does not influence termination (i.e. when $b \to \infty$), there is indeed a negative impact of cost avoidance and reward maximization, similar to [1]. (see all refrences below)
>
> ### 3. Re bounded rewards:
>
> Bounded rewards is a very common assumption in the RL literature [2]. Particularly, this assumption only scales the regret by a factor $R_{max}$, yet it helps maintain brevity. It is therefore convenient to set the rewards to the interval $[0,1]$. We note that, similarly, we can work with any non-negative sub-Gaussian rewards.\
> Finally, in our setting, this choice is orthogonal to the choice of costs. Specifically, the costs only affect the termination probability, and therefore a fixed cost function would have the same exact effect for any scaling of the reward function. This is due to the fact that the costs affect the obtained reward only implicitly through the transition function.
>
> ### 4. Re intuition on the theory, $\kappa$, $L$, and practical implications:
>
> To answer your question, we start with some intuition regarding $\kappa$. Suppose the costs are all very small and the bias is large, such that we are on the far left side of the sigmoid function. In this case we will get terminated only after a large number of steps, and the credit assignment problem (for the costs) will be very hard. In this case, estimating the costs becomes a hard problem, and this directly affects our regret. On the other hand, assume that our bias is small but costs are large, so that we are on the far right size of the sigmoid function. In this case, we will most likely be terminated after one step, but estimating the costs of different actions will be hard. This small gap will be evident in the regret after enough episodes. \
> Theoretically, the hardness stems from the logistic bandit problem (see for example [3], and particularly Proposition 4). There, $\kappa$ is inherent to the difficulty in estimating the parameters of the logistic model.
>
> Regarding your concern of practical implications. We believe that in practical applications the costs would be quite sparse, making the potential problem of a wide range of outcomes less apparent. Moreover, recall that in practice we use limited memory “windows” that mitigate the credit assignment problem. For example, in a recommender system, a user may exit an application due to bad recent recommendations, but is less likely to be affected by very old recommendations. \
> Accounting for longer horizons can be potentially done with a hierarchical approach,. Lastly, we emphasize that $\kappa$ is a real factor which exists in lower bounds of the estimation problem, as shown in [3].
>
> Regarding $L$ – this is a parameter that is commonly used in linear bandit papers [4,5], and provides a bound on the parameters. In our case, it bounds the parameters $c$, such that in the worst case $L \leq c_{\text{max}}\sqrt{SAH}$. In practice, $c$ is usually sparse, making this factor much smaller.
>
> We will clarify these points further by adding relevant discussion.
>
> ### 5. Re large windows:
>
> As the reviewer mentioned, using a larger window produced somewhat improved results. This overparameterization was good for the agent to improve learning as convergence of the costs was fast. We believe that in practical non-tabular settings, using a larger window can benefit the expressivity of the learned cost function. This allows the agent to travel through “wrong” cost functions, which can help it improve.
>
>
> ### Re Safety:
>
> We agree that this is an important point. The agent may find a way to “stop” the terminator through unsafe behavior. To avoid such bad outcomes, we should definitely take such scenarios into account and add safety constraints wherever possible. Particularly, termination does not eliminate the need for safe RL, but only complements it. Thank you for pointing this out; we will emphasize this in our work.

---

> > ### Author Response · Authors · 2022-08-02
> > **References**
> >
> > ### References:
> >
> > [1] Ross, Sheldon M. "Average cost semi-Markov decision processes." Journal of Applied Probability 7, no. 3 (1970): 649-656. \
> > [2] Auer, Peter, Thomas Jaksch, and Ronald Ortner. "Near-optimal regret bounds for reinforcement learning." Advances in neural information processing systems 21 (2008). \
> > [3] Abeille, Marc, Louis Faury, and Clément Calauzènes. "Instance-wise minimax-optimal algorithms for logistic bandits." In International Conference on Artificial Intelligence and Statistics, pp. 3691-3699. PMLR, 2021. \
> > [4] Abbasi-Yadkori, Yasin, Dávid Pál, and Csaba Szepesvári. "Improved algorithms for linear stochastic bandits." Advances in neural information processing systems 24 (2011). \
> > [5] Chatterji, Niladri, Aldo Pacchiano, Peter Bartlett, and Michael Jordan. "On the theory of reinforcement learning with once-per-episode feedback." Advances in Neural Information Processing Systems 34 (2021): 3401-3412. \

---

> > > ### Comment · Reviewer_nYZQ · 2022-08-07
> > > **Post-rebuttal response**
> > >
> > > Thanks to the authors for their response and clarifications. I've read through all the other reviews and responses, and am satisfied to recommend acceptance. I've adjusted my score upwards to reflect this

---

### Official Review · Reviewer_tpfd · 2022-07-10

**Rating:** 7
**Confidence:** 4
**Soundness:** 3 good
**Presentation:** 4 excellent
**Contribution:** 4 excellent

**Summary:**

The paper introduces an extension of the MDP paradigm, to the case where episodes can be interrupted by an external non-Markovian signal.  This has several practical applications for cases with human supervision. The paper includes the formalization of this setting, theoretical properties on provably-efficient learning, and practical algorithm that is evaluated on driving and MinAtar benchmarks.

**Questions:**

-	What is “L” in Sec.3, e.g. Thm 1 (l.123)?  This seems an important (cubic) factor in the bound, yet I could not see it defined.
-	How much is the ensemble needed?  This seemed somewhat of a distraction on the proposed method.  Its effect is investigated in the Ablation, and it seems to help. But would it be possible to achieve similar effect without the extra cost of an Ensemble?
-	What metric do you report specifically?  “We report mean and std” (l.214). Is this the std dev or the std error?  I suspect std dev, based on caption of Table 1. I would recommend std error to draw conclusions on significance, and verify whether you ran enough seeds.


**Limitations:**

-	The paper provides a reasonable discussion of technical limitations.
-	The paper provides some discussion of societal impact in the final discussion, at a level that is adequate for the potential risks.


**Strengths And Weaknesses:**

(+) The proposed setting is interesting, and I can indeed see many useful applications of this.  As such, this is a significant novel contribution.
(+) The paper is clearly written, and the reader can understand the methodology and contributions.
(+) Several experimental domains are considered, and ablations are performed to clarify impact of different aspects of the model.

(-) The paper makes a very specific assumption about the conditions under which termination happens (based on sum of costs), which seems more out of mathematical convenience than out of real-world motivation. This limitation is clearly stated upfront, and acknowledged in the discussion.
(-) More experiments would be helpful to draw stronger conclusions. Only 5 seeds are used (Fig.5 caption), yet there has been many reports that this is insufficient for high-confidence conclusions in RL (e.g. Henderson et al., AAAI 2018). Why not run 10 seeds? This should be feasible.
(-) The number of terminations needed to learn seem very high.  Can you report how many terminations are observed for each result?  The figures should # steps, so it’s not obvious to know how many terminations were observed.  But #steps is in the 10^6-10^7, which suggestions #terminations would be much higher than is practical for humans to give.  The fact that you had to train a termination for BDr Expt 2 highlights this limitation; it would have been preferable to show results with direct human termination, to be verify whether the proposed work meets the problem as stated.

---

> ### Author Response · Authors · 2022-08-02
> **Response**
>
> Thank you for your helpful review. We are encouraged that you found our work significant and novel, the paper easy to read, and that the experiments clarify the impact of the paper. Please find our response to your comments below.
>
> ### Re chosen termination model:
> While the function class of the termination model we describe is simple, we argue that it can capture expressive features as it depends on full trajectory states and actions. This is also apparent in the inverse RL literature, where a sum of rewards is assumed for the underlying human data. We believe this model can capture complex termination scenarios, such as the human termination example in our work.
>
> Your comment is also related to the question of using more complex function classes for termination. This is an interesting direction for future work. Particularly, for a cost function $P(\text{termination}) = f(\mathrm{c})$, where $\mathrm{c} = (c(s_1, a_1, c(s_2, a_2), \ldots, c(s_H, a_H))$, and $f \in \mathcal{F}$ is some function class, one would require to establish a result that obtains local confidence bounds for the costs, similar to Theorem 1. Particularly, the fact that we can establish local confidence bounds motivates our choice of model, as this lets us obtain an efficient algorithm for learning the costs and solving the TerMDP.
>
> Our practical solution can easily be adapted to the general class setting, by changing the sum over cost networks to some other function of the cost networks (see Figure 2), possibly a neural network. This is an interesting direction for future work - we will add further discussion on this to the paper.
>
> ### Re std and more seeds:
> We’ve run experiments of 5 new seeds and changed the std to corresponding 95% confidence intervals. This change did not have a meaningful effect on our results. We will upload an update of the paper with new plots.
>
> ### Re needed amount of termination signals:
> We agree that adding the termination-to-episode ratio is an insightful statistic, as you suggested. We gathered the data into the following table (which we will add to the paper):
>
> |                            |    **Backseat** | **Driver** |                |  **Minatar** |              |             |
> |:--------------------------:|----------------:|------------|:--------------:|-------------:|:------------:|:-----------:|
> |       **Experiment**       | **Coin Avoid.** |  **Human** | **Space Inv.** | **Seaquest** | **Breakout** | **Asterix** |
> |     **Number Episdoes**    | $6e6$           |    $6e6$   |     $10e6$     |    $10e6$    |    $10e6$    |    $10e6$   |
> | **Number of Terminations** | $0.15e6 \pm 0.04e6$       |  $0.14e6 \pm 0.03e6$ |    $0.26e6 \pm 0.14e6$   |   $0.3e6 \pm 0.08e6$  |   $0.28e6 \pm 0.05e6$  |  $0.34e6 \pm 0.05e6$  |
>
>
> We also agree that in the worst case, the number of required terminations can be large, and would theoretically scale as $O(T/H)$ where $T$ is the number of iterations and $H$ is the horizon. Still, we note that:
>
> - We are generally limited by theoretical lower bounds in RL, which require a large  iterations to converge (regret scales as $O(\sqrt{HAST})$). It is thus reasonable to assume that learning with implicit termination signals would also require large amounts of iterations to converge, unless the cost function is very sparse or simple. Nevertheless, as we show in Figure 6 (page 24, Appendix E), the cost function for Backseat Driver converges faster than the reward itself.
>
> - We emphasize that we do not necessarily control the termination signals. Particularly, we assume they are given to us as part of the environment. Our framework is designed to cope with this particular problem. Therefore, we do not necessarily need to think of terminations as a means for designing an algorithm, but as a constraint that already exists in the world, one which must be taken into account.
>
> ### Re definition of L:
>
> We define $L$ in line 84. We will remind the reader of this definition when it is used again. The parameter $L$ is an upper bound on the values of $c$. We note that this factor can scale in the worst case as $c_{\text{max}}\sqrt{SAH}$, if all costs are equal to $c_{\text{max}}$, though in practice can be much smaller if the cost function is sparse (which is usually the case). This factor is common in linear bandit [1,2] and logistic bandit literature, which we utilize for our confidence result.

---

> > ### Author Response · Authors · 2022-08-02
> > **Response (second part)**
> >
> > ### Re no optimism.
> > This is a great question. We first note that optimistic estimation of the costs is required by our theoretical analysis. A very common technique in RL for optimistic estimation is to use an ensemble [3,4], and therefore we chose this practice for our implementation. Indeed, as our experiments suggest, using this mechanism improved performance in all of the environments. We believe that a possible way to make this mechanism more efficient would be to use more workers, and we encourage it whenever it is computationally possible. We also note that, though we achieved optimism through an ensemble of cost networks, other methods of uncertainty estimation could be used (e.g., MC Dropout). Finally, our experiments suggest that even without the ensemble, our method still exceeds the baselines. Thus, the size of the ensemble might serve as a hyper-parameter which balances the performance and computational efficiency of the algorithm.
> >
> > ### References:
> >
> > [1] Abbasi-Yadkori, Yasin, Dávid Pál, and Csaba Szepesvári. "Improved algorithms for linear stochastic bandits." Advances in neural information processing systems 24 (2011). \
> > [2] Chatterji, Niladri, Aldo Pacchiano, Peter Bartlett, and Michael Jordan. "On the theory of reinforcement learning with once-per-episode feedback." Advances in Neural Information Processing Systems 34 (2021): 3401-3412. \
> > [3] Yu, Tianhe, Garrett Thomas, Lantao Yu, Stefano Ermon, James Y. Zou, Sergey Levine, Chelsea Finn, and Tengyu Ma. "Mopo: Model-based offline policy optimization." Advances in Neural Information Processing Systems 33 (2020): 14129-14142. \
> > [4] Peer, Oren, Chen Tessler, Nadav Merlis, and Ron Meir. "Ensemble bootstrapping for Q-Learning." In International Conference on Machine Learning, pp. 8454-8463. PMLR, 2021.

---

> ### Comment · Reviewer_tpfd · 2022-08-08
> **Post-Rebuttal Response**
>
> I thank the authors for their detailed response.  I encourage you to make the changes & clarifications discussed.  I recommend acceptance of the paper.

---

### Official Review · Reviewer_gYNG · 2022-07-11

**Rating:** 7
**Confidence:** 3
**Soundness:** 3 good
**Presentation:** 3 good
**Contribution:** 3 good

**Summary:**

This paper studies the RL setting with external / forced termination and defines a new type of MDP, the Termination MDP or TerMDP. The authors propose a theoretically-grounded method which is well suited to this setting and derive regret bounds. They also introduce a scalable approach, TermPG, which combines optimism and a dynamic discount factor incorporating. The method is evaluated on two different domains, including an autonomous driving scenario with human termination data. TermPG is significantly better than vanilla PG and other variants.


**Questions:**

1. In the introduction, you mention that the cost function is state-dependant, but in the first equation for the termination probability, it looks like it is a function of state-action pairs. Could you write it as a state-only function? Would this change the formulation, analysis, algorithm, or results?

2. Why did you choose this particular probability as a function of the termination? Could this be modelled in a different way or what are the advantages / disadvantages of this choice?

3. The TerMDP formulation seems to share some similarities with the IRL (inverse RL) paradigm where the agent receives some reward but it is not the full reward (there is uncertainty, some unknown part of the reward which needs to be inferred from the environment -- in this case the termination actions). Could you comment on this?

4. In the related work you mention several relevant areas such as contrained MDPs, reward design, global feedback, or preference-based RL. However, you don't compare your methods with any of those (or some variants adapted to this setting). Could you add such baselines or at least explain why those aren't being considered?


**Limitations:**

The paper addresses limitations and potential negative social impacts in the last section. I found this section to have a reasonable amount of detail and nuance. However, it would still be good if the authors can further expand (perhaps in the appendix) on how this approach compares with others, particularly different types of reward shaping, inverse RL methods.

**Strengths And Weaknesses:**

## Clarity

The paper is generally clear and well-written.


## Soundness

The introduced setting and method are well-motivated with practical examples, analyzed through a theoretical lens, and thoroughly evaluated on a number of challenging tasks. The authors compare with a number of ablations, demonstrating the importance of each design choice. The proposed approach seems significantly better than naive algorithms in this setting, across many environments.


## Novelty and Significance

The paper introduces a new type of MDP, the TerMDP, together with a method suited for this setting TermPG, which is novel as far as I know. I particularly liked the optimistic discounting factor part of the algorithm. The paper also introduces a new benchmark and evaluation protocol in a realistic domain for evaluating methods for TerMDP. I also appreciated the fact that the authors ran experiments with human data, rather than merely synthetic data or human proxies (which are typically quite disconnected from reality). It is promising that the results are strong in those cases. While this setting is quite niche / specific and I wouldn't expect a large part of the community to build on this work, I believe it is still important for certain real-world applications and for building systems that won't act in isolation but rather interact with humans, so I hope it will inspire more research in this area. The open sourcing of the code for environments and methods could help advance research in this area, so I encourage authors to make these publicly available.

---

> ### Author Response · Authors · 2022-08-02
> **Response**
>
> Thank you for your positive review and your helpful comments. Please find answers to your questions below:
>
> ### 1. Re State-action dependent cost vs state-dependent cost:
> Using state-only costs would not change the analysis. We can write the termination probability using a state-only cost function, which would be a special case of the state-action cost function. This would relate to the setting in which every action has the same cost. To make this clear, we’ll rephrase the wording of state-dependent cost function to state-action-dependent cost function.
>
> ### 2. Re Different function classes for termination:
> Thank you for this important question. The case of more complex function classes is indeed interesting. For a cost function $P(\text{termination}) = f(\mathrm{c})$, where $\mathrm{c} = (c(s_1, a_1), c(s_2, a_2), \ldots, c(s_H, a_H))$, and $f \in \mathcal{F}$ is some function class, one would need to establish a result which obtains confidence bounds over the costs, similar to Theorem 1. Particularly, local confidence bounds are essential for learning the costs and solving the TerMDP.  The simplicity of our model enables derivation of such a high-confidence result, while at the same time we found it to be expressive enough in practice. Moreover, our practical solution can easily be adapted to the general non-linear class setting, by changing the sum of cost networks to some other function of the cost networks (see Figure 2), possibly a neural network. This is an interesting direction for future work, and we hope it would spark further research. We will add more discussion on this matter to the paper.
>
> ### 3. Re similarities to IRL:
> We agree that the IRL problem resembles the cost estimation problem. However, in the former, one must recover the unknown reward function assuming optimality, while here one must recover an unknown cost function from sparse termination signals. An important difference between these is the role of the cost function, in contrast to the reward. While the reward is optimized directly, the costs in the TerMDP affect the transition probability to a terminal state. This makes this inverse problem quite different. Also, the reward and cost functions may play different roles in the TerMDP setting, as the terminator’s preferences may not align with the designed reward function.
>
> Another way to view this problem is as an inverse problem for a state dependent discount factor. One could attempt to construct an IRL problem in which both the discount (= 1 - probability of termination) and the reward function must be recovered.
>
> ### 4. Re relation to reward design and preference-based RL:
> The TerMDP model considers a new method to adapt to external feedback. There are two main scenarios to take into consideration with termination:
>
> a. Termination by an exogenous observer - this is the setting we mostly discuss in the paper, which assumes we do not control the termination signals and they are given to us as part of the environment. Our framework is designed to cope with this particular problem. Therefore, comparison to reward design / preference-based methods is not very informative in this setting, as we consider a different feedback mechanism.
>
> b. Termination as a design choice - we can view termination as an event we trigger ourselves for improving RL algorithms. This would better relate to the preference-based / reward design setting. Reward design is quite a hard problem, and it is unclear how to systematically accomplish this task. Preference-based methods are great for incorporating external knowledge without the need to design a reward function. Termination can be considered as an even easier alternative to incorporate such knowledge - as termination signals are very sparse and require very little information from the external observer. This was evident in our experiments (Section 5, lines 241-250), where we gave the human observers a very general objective for terminating the agent, which they were able to easily interpret. \
> As an example, consider the case where termination happens when the agent “gets lost” or approaches something known to be suboptimal. In that case, termination may signal the RL system that it had ventured to undesired territory. In some applications, designing such a termination signal may be a lot easier than designing a reward function that may percolate to the desired area of the state space.
>
> Finally, we note that the cost function need not be related to the reward function, in the sense that the terminator may have preferences that are not aligned with our primary objective, in contrast to preference based / reward shaping methods.
>
> We’ll add further discussion on the relation of these problems to our work.
>
> -----------------------------------------------------------------------------------------------
> We also note, as the reviewer mentioned, that we will open-source all our code and environments.

---

> > ### Comment · Reviewer_gYNG · 2022-08-06
> > **Post-Rebuttal Response**
> >
> > Thank you for the clarifications. After reading the other reviews and the authors' responses, I keep my score and recommend the paper for acceptance. I encourage the authors to udpate the paper with the additional discussions and clarifications suggested by the reviewers.

---

### Author Response · Authors · 2022-08-02
**To all reviewers**

We thank the reviewers for spending the time and effort to carefully evaluate our work. We are encouraged that the reviewers found our work “novel” [gYNG, tpfd], “extremely well written” [nYZQ], and “well motivated” [nYZQ] with “many useful applications” [tpfd] and “real-world applications” [gYNG].  Beyond these encouraging descriptions, the reviewers also made valuable comments that we answer in the following.

---

### Meta-Review · Area_Chair_myAw · 2022-08-26

**Recommendation:** Accept
**Confidence:** Certain

**Metareview:**

All reviewers are in agreement that this paper should be accepted. It combines clear writing, a well-motivated setting (external termination due to unobserved accumulation of costs), and sound theoretical analysis with a novel algorithmic contribution (TermPG) that performs well on an interesting domain that aligns well with the stated setting. Furthermore, the additional leveraging of the cost estimation for dynamic discounting may itself be of fairly broad interest to research in RL. Clear Accept, really solid paper.

**Award:**

No

---

### Decision · Program_Chairs · 2022-09-14

Accept